# Water and Us: tales and hands-on laboratories to educate on sustainable and nonconflictual water resources management

Francesca Munerol[1,*], Francesco Avanzi[1,*], Eleonora Panizza[1], Marco Altamura[1], Simone Gabellani[1], Lara Polo[1], Marina Mantini[1], Barbara Alessandri[1], and Luca Ferraris[1]

[1]CIMA Research Foundation, Via Armando Magliotto 2, 17100 Savona, Italy
[*]These authors contributed equally to this work.

**Correspondence:** Francesca Munerol (francesca.munerol@cimafoundation.org) and Francesco Avanzi (francesco.avanzi@cimafoundation.org)

**Abstract.**

Climate change and water security are among the grand challenges of the 21st century, but literacy on these matters among high-school students is often unsystematic and/or detached from the real world. This study aims at introducing the educational objectives, methods, and early results of "Water and Us", a three-module initiative that can contribute to advancing water education in a warming climate by focusing on the natural and anthropogenic water cycle, climate change, and emerging water conflicts. The method of Water and Us revolves around storytelling to aid understanding and generate new knowledge, learning by doing, a flipped classroom environment, and a constant link to examples from the real world – such as ongoing droughts across the world or seeds of conflicts around transnational river basins. Water and Us was established in 2021-2022, and during that school year involved 200+ students in a proof of concept to test the complete didactic approach in small-scale experiments. Results from 40+ hours of proof-of-concept events preliminarily confirmed the effectiveness of this approach in conveying the essential elements of the natural and anthropogenic water cycle, the most recurring concepts related to climate change and water, and possible conflicts and solutions related to water scarcity in a warming climate. The Water and Us team remains interested in networking with colleagues and potential recipients to scale up and further develop this work.

## 1  Introduction

Climate change is the big elephant in the room of our times. Fueled by anthropogenic emissions of greenhouse gases, climate change "has caused widespread adverse impacts and related losses and damages to nature and people, beyond natural climate variability" (IPCC, 2022). These impacts include an increase in heatwaves and extreme precipitation, an increase in human and tree mortality, wildfires, ocean acidification and sea level rise, damages to ecosystems, and reduced food security (IPCC, 2022). While some steps have been taken since the seminal United Nations Framework Convention on Climate Change in Rio de Janeiro (1992), and while attention from the public has increased thanks to initiatives like Fridays for Future, the United Nations Sustainable Development Goals, or the Sendai Framework for Disaster Risk Reduction, "most observed adaptation is fragmented, small in scale, incremental, sector-specific, designed to respond to current impacts or near-term risks, and focused more on planning rather than implementation" (IPCC, 2022). Current consensus, both at scientific and societal level, is that

challenges related to climate change mitigation and adaptation will characterize the world for several generations to come
(Hansen et al., 2013; Zhenmin and Espinosa, 2019).

If climate change is the elephant, then the most proximal resource to humans and ecosystems – water – is the floor over which the elephant is standing. A rise in temperature as predicted by climate-change scenarios will lead to an increase in drought episodes (Spinoni et al., 2018), a rise in extreme events (Alfieri et al., 2017), a decline in snow water resources (Mote et al., 2018; Musselman et al., 2021), glacier depletion (Shannon et al., 2019), an imbalance between water demand and availability (Barnett et al., 2005; Immerzeel et al., 2020), and ultimately profound alterations in the whole water cycle (IPCC, 2022). Given the extent and intensity of human water management, such changes may ultimately result into societal instability, conflicts, poverty, displacement, and less water security at global scale (Kelley et al., 2015; Galli et al., 2022). This is particularly true where precipitation is highly seasonal and/or where snow and glaciers play a fundamental role in storing water during wet and cold winters to release it during warm and dry summers (Barnett et al., 2005; Avanzi et al., 2023). Such changes in the water cycle will inevitably challenge our societies, since the water cycle, ecosystems, and human societies are and always will be intimately connected.

Despite these intimate connections, contemporary geosciences and, by reflection, water education from elementary to high schools often remain anchored in a traditional view of the water cycle as a physical process where humans have little to no role. Meanwhile, surveys from various parts of the world show that students tend to confound mitigation and adaptation to climate change with unrelated environmental issues (Bofferding and Kloser, 2015), while knowledge gained at school often does not translate in everyday habits (Amahmid et al., 2019) due to current high school students' environmental literacy possibly being inadequate (Wardani et al., 2018). These experiences speak for a need to expand how climate change education is done (Harker-Schuch and Bugge-Henriksen, 2013), as acknowledged by UNESCO (https://en.unesco.org/themes/water-security/hydrology/water-education, last access 04/09/2022).

This paper aims at introducing the objective, methods, and early results of "Water and Us", an educational initiative developed by CIMA Research Foundation (Italy) to encourage scientists and teachers to co-deliver lectures on three topics: the natural and anthropogenic water cycle, climate change, and emerging water conflicts. The initiative thus contributes to filling knowledge gaps on the important, but often poorly understood link between water resources and security, climate change, and institutional governance. Water and Us is an interdisciplinary initiative bringing together hydrologists, jurists, and communication experts, and is strongly committed to cross borders within and across scientific fields of study to modernize water education in a warming climate.

The paper is organized as follows: Section 2 introduces the educational approach of Water and Us, with particular emphasis on its objectives, their implementation into three modules for high schools, and their counterparts for elementary schools and adults. Section 3 discusses an array of metrics we are identifying to evaluate this approach in the context of our early results for this initiative. Finally, Section 4 draws conclusions.

## 2 The educational approach

In describing the current method of Water and Us, we will inevitably refer to our specific experiences in Italy – see the example of storytelling in the Appendix regarding the Alpine 2022 drought. However, the approach described here is fully transferable, with Chapter 4 of the IPCC Assessment Report 6 on Water providing a general framework to identify emerging risks for a given location and how these risks link to governance challenges (see https://www.ipcc.ch/report/ar6/wg2/downloads/report/ IPCC_AR6_WGII_Chapter04.pdf, last access 08/05/2023). The authors are also initiating an open, online repository of the materials used (https://doi.org/10.5281/zenodo.8341482). The only requirement is for students to have a basic understanding of the water cycle.

The primary target audience of Water and Us is high-school students (in Italy, 14 to 19 years old), but the offer has already been adapted for elementary schools and adults. The choice of high-school students was due to two main factors. The first is the clear fit between this initiative and high-school programs in Italy. In this regard, Water and Us directly contributes to civics (in Italian, Educazione Civica, see https://www.istruzione.it/educazione_civica/, last access 13/09/2023), which includes educational targets on sustainability and environment awareness, and to science programs, which include chapters on the Earth system, the water cycle, and climate. The second is the symbolic leverage of high schools representing the last step of mandatory education in Italy, which means that high-school students are in the process of deciding their own future when they are exposed to Water and Us, an aspect that promotes engagement and awareness.

The main structure of the initiative revolves around three objectives, four didactic pillars, and three modules (each needing 1.5 to 2 hours). Figure 1 summarizes these pillars and the content of each module.

### 2.1 Objectives

The main educational objectives of Water and Us are as follows:

1. inform next generations on the concept of "water resource" as an intertwined result of the natural water cycle and anthropogenic actions, and on how, where, when, and by whom water is used, transported, stored, and diverted in the Anthropocene;

2. educate students on the most salient aspects of climate change and its governance, including the difference between mitigation and adaptation, the role of international agreements, the scientific foundation of global warming, and how these processes affect water availability at all scales – including future scenarios of water supply, floods, and droughts;

3. raise awareness on existing and potential governance conflicts around the use of water, especially in a warming climate, and on solutions for a nonconflictual water resources management.

These objectives are well nested into the Sustainable Development Goals, and in particular #4 (Ensure inclusive and equitable quality education and promote lifelong learning opportunities for all – Target 4.7 on ensuring that all learners acquire the knowledge and skills needed to promote sustainable development), #10 (Reduce inequality within and among countries –

Target 10.2 on empowering and promoting the social, economic and political inclusion of all), and #13 (Take urgent action to combat climate change and its impacts – Target 13.1 on strengthening resilience and adaptive capacity to climate-related hazards and natural disasters in all countries).

## 2.2   Didactic pillars

From a methodological standpoint, Water and Us leans on four overarching pillars (Figure 1).

The first is an educational approach based on storytelling, under the assumption that the ancestral attraction of humans towards tales will gain their attention and enhance understanding. This first pillar goes well beyond "telling anecdotes": instead, it nests itself in a broad body of empirical and theoretical literature in education showing that storytelling can significantly reduce depersonalization, develop identities, promote empathy and diversity, aid with understanding of complex issues by linking them to the proximal world experienced by students, and ultimately generate new knowledge (Abrahamson, 1998; Collins, 1999; Haigh and Hardy, 2011; Hibbin, 2016; Astiz, 2020). In doing so, Water and Us seconds the advent of digital devices and so digital storytelling to generate vivid experiences for students through the mixture of voices, images, and videos (Robin, 2008). Note that our stories focus on contemporary events, such as ongoing droughts across the world, rather than traditional tales (see the Appendix for an example).

The second pillar are hands-on experiences, to immediately put theory learned into practice. During our events, for example, students are asked to identify potential water stakeholders in familiar and less familiar landscapes, and then to impersonate these stakeholders in focus groups to reflect upon their needs with regard to water and how these needs may conflict with (or be in synergy with) other stakeholders. Groups are finally asked to discuss these findings in an effort to tackle emerging conflicts and maximize synergies (see Section 2.3 and 2.5). Thus, we openly link Water and Us to the long-standing educational tradition of "learn by doing" (Schank, 1995) to go beyond the artificial setting of school education and allow for a more natural, immediate understanding of the subject matter.

Learning by doing is connected to the third pillar, which is a flipped classroom environment in which students become the protagonists of the teaching experience. To this end, each module in Water and Us includes workshops led by the students for the students. In this framework, storytelling introduces the minimum amount of knowledge required by students to conduct the workshops themselves (lack of preparation being a frequent problem with flipped classrooms, Akçayır and Akçayır, 2018). While relatively new, this flipped classroom approach has already been widely applied, with proven benefits (Awidi and Paynter, 2019).

The fourth pillar is a constant link to the real world, and in particular to the most pressing, contemporary societal issues – water security and climate change. The hypothesis here is that focusing on the real world will make topics covered by Water and Us more tangible and so more interesting to students, as they can directly relate to their future in a climate-change 21st century. This is in line with existing literature showing that climate-change education must be accessible and action-oriented (for example, see Lee et al., 2013). In this regard, Water and Us synthesizes a geoscience-based approach to climate change with policy and governance, in an effort to make this initiative open to all aspects of water in the modern era.

**PILLARS**

| | MODULE 1: READ THE WATERSCAPE | MODULE 2: THE 21st CENTURY TOOLBOX | MODULE 3: WATER CONFLICTS |
|---|---|---|---|
| **STORY-TELLING** | Parallelism between the Californian 2012-2016 drought and the Italian 2022 drought | | Parallelism between the water crisis in Lake Turkana and in Lake Maggiore + 2022 italian drought |
| **LEARN BY DOING** | Students, gathered in groups, learn how to identify who uses water, how, and why | In groups, students search for the meaning of an assigned list of climate-related words, alongside accredited sources | By role gaming, students work in groups to better understand how stakeholders act based on their needs |
| **FLIPPED CLASSROOM** | Students then report their findings to the class | Groups then exchange, discuss and negotiate definitions and sources | Students, role playing as the stakeholders, report their water needs and strategic positions to the other groups |
| **REAL WORLD** | The Californian and Italian droughts | Climate change, IPCC, Paris Agreement etc. | Lake Turkana & Lake Maggiore, water conflicts |
| **MAIN POINTS** | -warmer temperatures cause changes in the water cycle<br>-humans affect the water cycle<br>-snow as a key reservoir | -we need an accurate vocabulary to describe 21st century climate challenges<br>-to master it means getting the chance to make an impact | -water conflicts exist and may exacerbate in the future<br>-new generations can be part of climate solutions |

HOW THE 4 DIDACTIC PILLARS COME INTO PLAY IN WATER AND US

**Figure 1.** The four overarching pillars of Water and Us (first column) and how they come into play in the three educational modules.

## 2.3 Module 1: read the waterscape

Starting from the four didactic pillars outlined above, the first module of Water and Us focuses on the water cycle in a warming climate (Figure 2). This module builds from the premise that water is an essential resource for life on our planet to make three broader points. First, that the natural water cycle of evaporation – precipitation - runoff is now part of a much broader and more complex mechanism including regulations, allocations, and demands by human societies, which can significantly change the natural course of water across our planet and introduce a striking variety of water stakeholders (Sivapalan et al., 2012). Second, that in temperate regions of the world this natural/anthropogenic water cycle relies on an intermediate natural reservoir, snow, which is often overlooked and rarely seen as a key precondition for life on our planet (Barnett et al., 2005). Third, that this natural/anthropogenic water cycle is changing, due to a recurring pattern in temperate regions of warmer temperatures, less snow, and eventually less available water (IPCC, 2022).

We originally chose to make these points by linking future scenarios of temperature, snow, and water supply in Italy with an exemplary story from another part of the world, the California 2012-16 snow drought (see Harpold et al., 2017, and the Appendix). By showing real-world implications of the link between warmer temperatures, less snow, and less water, the California

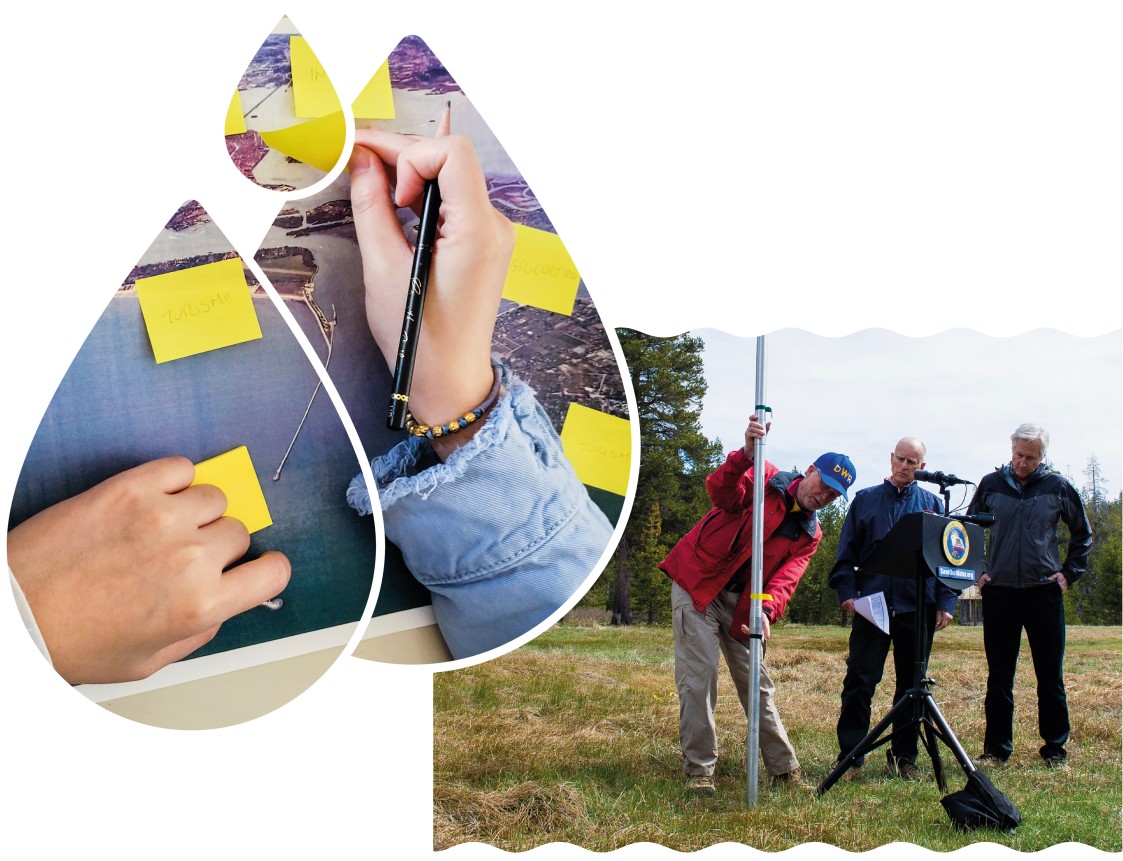

**Figure 2.** Some content from Module 1 of Water and Us: students learning how to read the waterscape (who is using water, where, and why?) and an iconic image of the California snow drought (then Governor Jerry Brown taking part in the 2015 Snow Survey at Phillips Station, the first with no snow on the ground in April– credits: CA Department of Water Resources).

drought is a perfect example of the challenges posed by global warming and increased aridity for the natural/anthropogenic water cycle we live in.

Our events started in January 2022 and so we soon had to re-adapt this framework to include the unfolding Italian drought (see the Appendix). We made the pragmatic choice of preserving the California story, but progressively included parallels to the 2022 Italian temperature and precipitation anomalies, snow deficit, and streamflow lows. We found that doing so enhanced credibility of our stories as students appreciated patterns across continents and were able to find links to topics that were covered by media and social networks at the time. Albeit unfortunate in nature, this coincidence of events made Water and Us

concrete and relevant to students.

    The focus on droughts was instrumental, as it allowed us to link Water and Us to our own experiences related to climate change and water and thus make communication more effective for our audience. We acknowledge that other water risks may

be relevant to different contexts, cultures, and representations of what is at stake, such as sea level rise (Cazenave et al., 2014), emerging flood pressure (Hirabayashi et al., 2013), shrinking glaciers endangering mountain communities (Council et al., 2012), or increasing desertification (Stringer et al., 2009). Even maintaining a focus on droughts, other episodes can be useful to contextualize local events in a global framework, such as the multi-year drought in the Andes (Rivera et al., 2017) or in Australia (Saft et al., 2015). The key ingredient of this first module is the focus on nature-human interactions around the use of water, and how these interactions are challenged in a warming climate.

The first module always ends with a workshop, dedicated to putting gained knowledge about the natural/anthropogenic water cycle into practice. We gather students in small groups (4 to 5 maximum members), assign one landscape to each of them (see an example in Figure 2 and others in our online repository at https://doi.org/10.5281/zenodo.8341482), and ask students to pinpoint who is using water, how, and why – students thus train themselves to read the waterscape. Students are left with approximately 15 minutes to accomplish this goal, and are asked to write their notes on sticky notes that are then placed on their waterscapes. At the end of this work, students report their findings to the class and come up with a bottom-up, shared categorization of recurring water stakeholders (see Section 3). We found this to be particularly important not only because knowledge of water stakeholders is a precondition to understand following modules in Water and Us (and more generally what is at stake regarding water security in a warming climate, see Section 2.5), but also because most high-school students we interacted with reported that the last time they were taught about the water cycle was in elementary school.

### 2.4 Module 2: the 21st-century toolbox

The second module focuses on climate change, a term that is well known to students but that – often – few are able to clearly explain. This gives us an opportunity to convey two main messages: 21st-century challenges have a precise and accurate vocabulary and handling this vocabulary is a precondition for next generations to play an impactful role in shaping the future. At the same time, information that can be gathered from current media can be inaccurate, or simply partial. This second module of Water and Us aims at going from such incomplete definitions to a coherent picture of ongoing climate-change debate at global scale.

Different from module 1 and 3, module 2 is entirely based on a workshop (Figure 3). Students are again shuffled into small groups and are assigned a list of terms related to climate change, comprising "global warming", "IPCC", "COP", "sustainability", "greenhouse gases", "mitigation and adaptation", "extremes", "drought", "floods", and "Paris Agreement". We then ask students to use their own knowledge and digital devices to come up with an accurate, and yet concise definition of each of these terms. While doing so, we ask them to check for multiple sources, note down these sources, and discuss how and why definitions may differ across them. After this first round, each group of students is asked to explain their definition to the class to not only improve shared knowledge, but also potentially compare definitions across groups and so realize the quality of accredited and independent sources.

This workshop can be iterative, based on available time and feedback from students: for example, we often notice a particular interest from students about IPCC, and so go through a second round with words like "RCP", "climate", or "future scenarios"

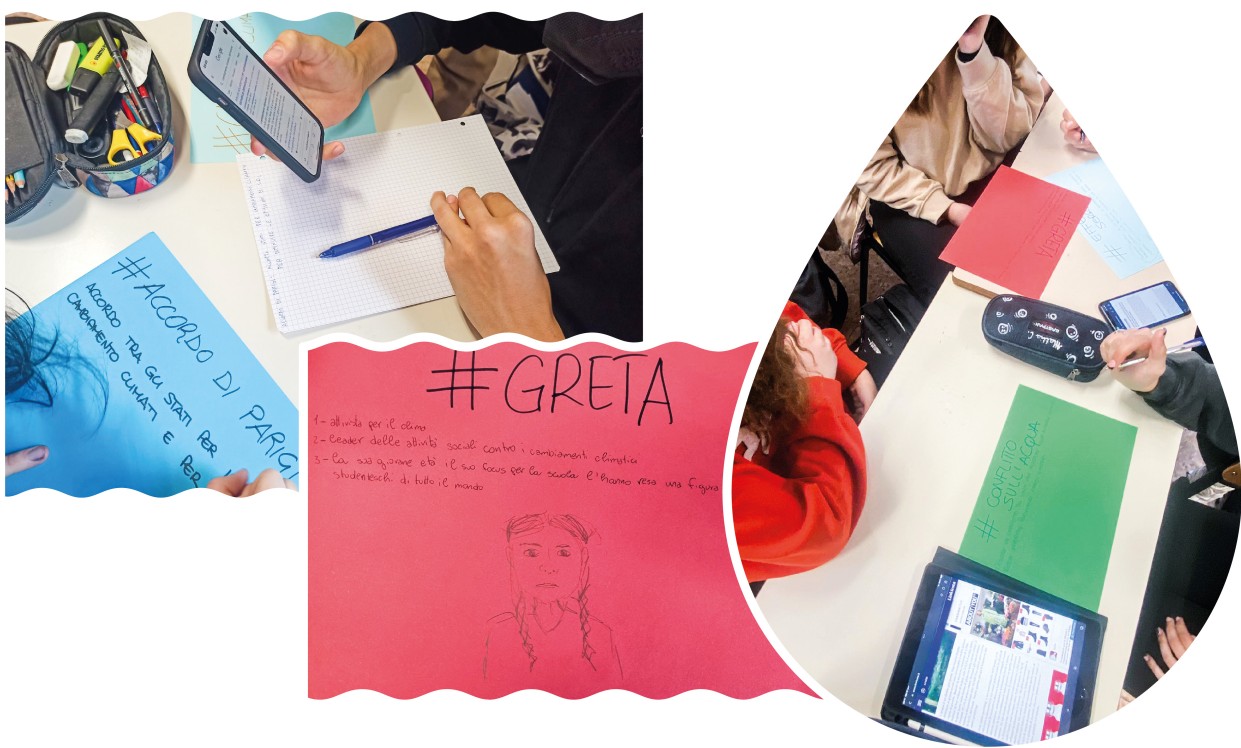

**Figure 3.** Some content from Module 2 of Water and Us: students proposing definitions of the most recurring terms related to climate change.

to second their interest in this sense. At the end of the workshop, we encourage students to note down the final definitions for them to keep a toolbox for future use.

### 2.5 Module 3: water conflicts

Module 3 connects the dots between the previous two modules and focuses on the main societal implication of a changing
climate in a natural/anthropogenic water cycle: emerging water conflicts.

Discussing water conflicts with Italian students may be challenging, because they tend to associate these matters with more arid regions of the world. To overcome this issue, we break down our story in two parts. The first is indeed quite exotic for our audiences and deals with the water crisis concerning Lake Turkana and how it is associated to climate variability (Yongo et al., 2010). For our audience, this has the classical setting of water crises as they expect them. We then move to a much less known
situation, the transnational management of Lake Maggiore across Italy and Switzerland and how it is exacerbated by ongoing climatic extremes (Guariso et al., 1985). We show how national resolutions on lake level have already led to court decisions

or tensions across stakeholders, and how these tensions are indeed seeds of potential, future water conflicts. Here again, the mounting 2022 Italian drought gave us an unfortunate opportunity to bring newspapers and media coverage to classes and discuss concrete examples of these seeds, such as public conversations on who was the priority water user, or how and when to divert water from one river to another for drought relief.

We end module 3 with a role game, where each group of students chooses one category of water stakeholders as they were identified during module 1 (the most recurring ones being farmers, industries, civil water supply, ecosystem conservation, hydropower, and tourism – see Section 3). Each group is first asked to reflect on their specific need concerning water (when and where do we need water? Why do we need it?) and what decisions they would like society to make in their own interest (e.g., some stakeholders may want little to no water restriction, while other stakeholders may be in favor of specific water infrastructures). After reporting these needs and positions to the other groups, they gather again to identify what are the strategic positions they can take to achieve their needs and what positions may, instead, represent a seed of conflict (e.g., some economic sectors may second ecosystem conservation, but may dislike priority allocation to other sectors, and so forth, Figure 4). This workshop ends by summarizing potential conflicts and synergies on a poster (see Figure 4 and Section 3), which remains to the class as another deliverable of Water and Us in addition to waterscapes and the climate-change vocabulary discussed in the previous sections.

## 2.6  Elementary students and adults

Adapting Water and Us to elementary schools (6-11 years old) required rethinking the structure and content of the program to identify a set of messages that were both effective to communicate to children and in line with the overall concept of this initiative. We thus selected three core messages related to the importance of water sustainability: first, water is the most precious resource on Earth, because we all need it to live; second, water must be preserved and not wasted; and third, we are not the only ones needing water.

From a methodological standpoint, the elementary edition of Water and Us consists of one module, of approximately 2 hours. We start with brainstorming around three questions: where does water live? What do we need water for? Who uses water? The answers to these questions are noted on the blackboard and remain visible throughout the event, as they will be used in a third step to reorganize students' knowledge of the natural and anthropogenic water cycle (Figure 5).

The second step involves telling a story to students while they look at iconic images drawn in color on large card-boards (see Figure 5). The story is about a child who becomes a friend of water through the typical "ups and downs" of child relationships: they initially enjoy playing together, but soon start to play pranks on each other (for example, the child wastes or pollutes water, while water takes revenge with flooding). At the end of the story, children become aware that water is their closest friend as it follows them in all aspects of their everyday life. The concept, structure, and development of the story are all geared towards getting students to relate to this story, while seeing parallels with their daily friendships. This story is available on our online repository at https://doi.org/10.5281/zenodo.8341482.

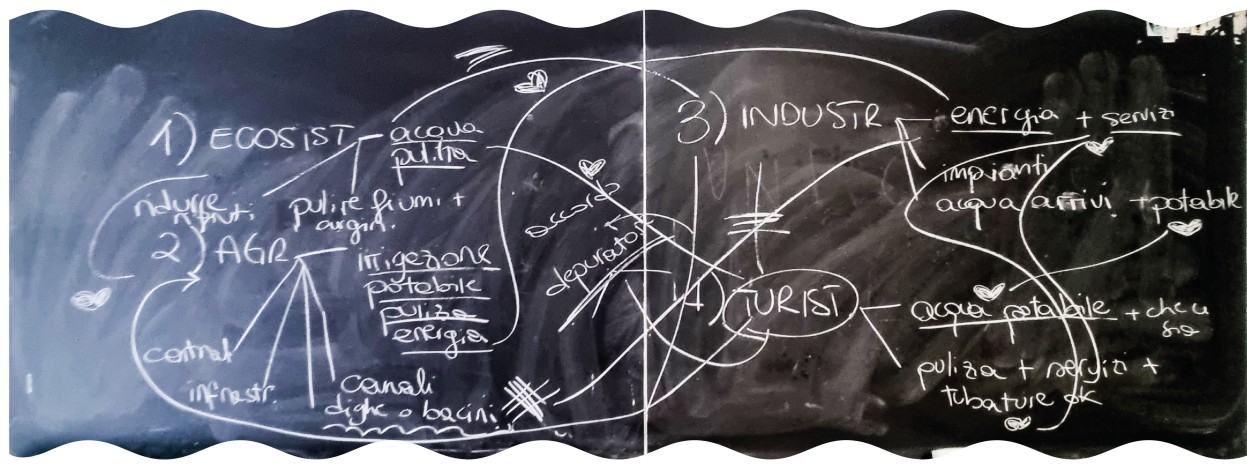

**Figure 4.** Some content from Module 3 of Water and Us: a map of water stakeholders (industries, ecosystems services, agriculture, tourism) and potential synergies-competitions.

The final step is to involve students in a drawing workshop: they represent situations in which they have been friends of water and, on the other side of the sheet, situations in which they have been enemies of water. In doing so, we stimulate causal discussions to get feedback and reinforce their learning of core messages.

Adapting Water and Us for adults is still work in progress. As of today, our main experience has been with traditional seminars or lectures to professionals or philanthropic organizations about water, climate change, and conflicts. Despite being more conventional in structure and development, we do preserve the central role of storytelling with adults. Here again, our experience is that starting from real-world stories, such as again the California drought or Lake Turkana, is an effective way of conveying upper-level concepts like climate change and sustainability.

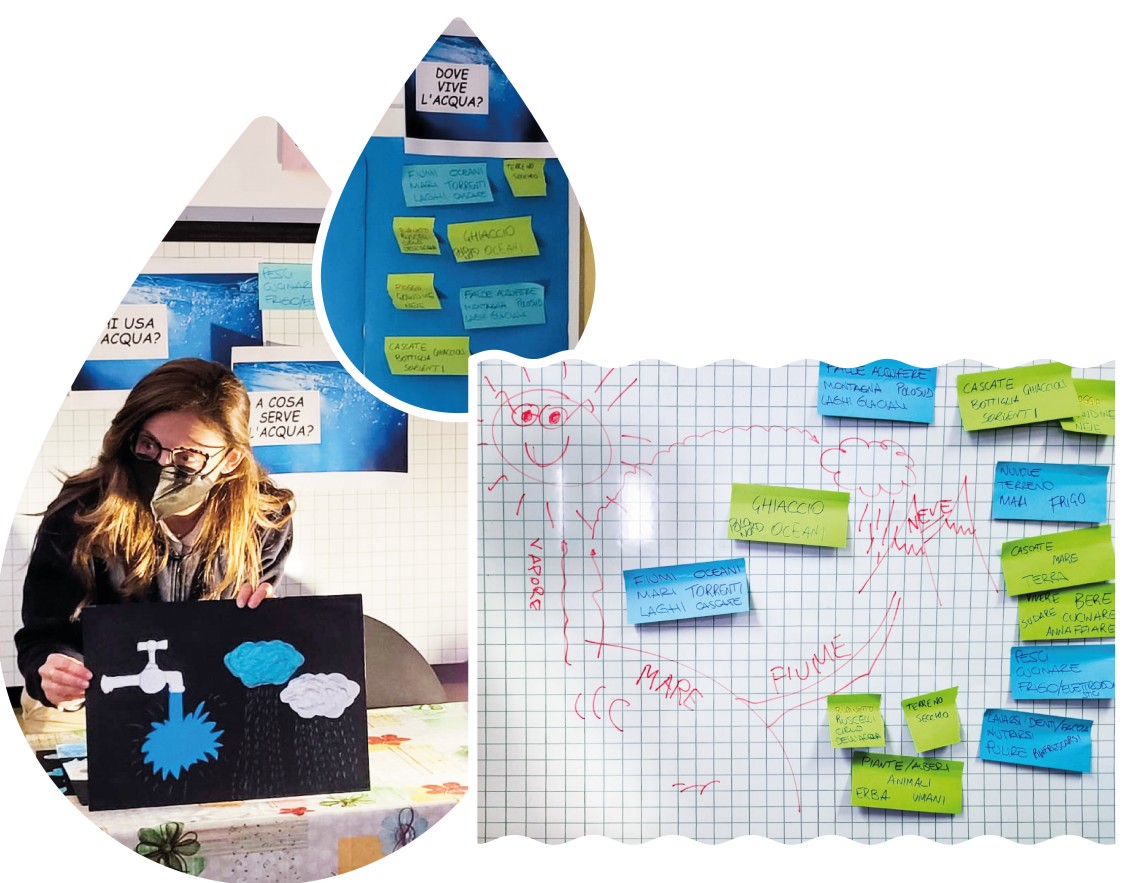

**Figure 5.** Snapshot from the elementary-school edition of Water and Us, where students are engaged through a dramatization of a fictional friendship between a child and Water.

## 3 Pathway to impact

Water and Us was established in autumn 2021 and worked as a proof of concept during the 2021-22 school year. The goal of this first phase was to develop the main portfolio of activities, to test it in the real world with small-scale experiments, and to leverage these experiences to identify indicators to validate the method and capture its impact. This phase involved 3 schools, 200+ students and 100 adults, and 40+ hours of events. In this section, we elaborate on these experiences and how they informed an array of proposed indicators (see Table 1).

Regarding objective #1 ("inform next generations on the concept of "water resource", as an intertwined result of the natural water cycle and anthropogenic actions, and on how, where, when, and by whom water is used, transported, stored, and diverted in the Anthropocene"), we propose using the average and variance of the number of water stakeholders identified on these waterscapes across groups as a concrete measure of the effectiveness of module 1 in communicating the complexity of the

| Objective | Indicator | Monitoring Method |
|---|---|---|
| Objective #1 | Average/variance # of water stakeholders identified on waterscapes | Sticky notes |
| | Average/variance # of identified seasonal data for water stakeholders on waterscapes | Sticky notes |
| Objective #2 | Average/variance # of students with pre/post awareness of climate change | Questionnaire |
| | Average/variance # of students with pre/post awareness of IPCC | Questionnaire |
| | Average/variance # of students with pre/post awareness of the Paris Agreement | Questionnaire |
| Objective #3 | Average/variance # of identified synergies across water stakeholders | Sticky notes |
| | Average/variance # of identified conflicts across water stakeholders | Sticky notes |
| | Average/variance # of identified "collaborators" across water stakeholders | Sticky notes |
| General impact | Number of students involved | Organizers' data record |
| | Number of teachers involved | Organizers' data record |
| | Number of schools | Organizers' data record |
| | Number of hours | Organizers' data record |
| | Percentage of audience in elementary schools | Organizers' data record |
| | Percentage of audience in high schools | Organizers' data record |
| | Percentage of audience in non-student positions | Organizers' data record |
| | Percentage of students taking part to all workshops | Organizers' data record |
| | Number of schools per year requesting new editions of Water and Us | Organizers' data record |
| | Number of schools per year requesting follow-up editions of Water and Us | Organizers' data record |
| | Number of PCTO programs associated with Water and Us | Organizers' data record |
| | Number of career-related follow-up questions by students | Organizers' data record |

**Table 1.** Proposed set of indicators to measure the impact of Water and Us. PCTO means "Paths towards Cross-cutting Skills and Orientation" (in Italian, Percorsi per le Competenze Trasversali e per l'Orientamento, see Section 3).

anthropogenic water cycle in the modern era. The average and variance of the number of temporal aspects that each group was able to attach to stakeholders (when do they need water?) could also be used to further shed light not only on water users, but also on how their needs intersect with each other in time.

In this regard, Table 2 summarizes the main stakeholders identified by students during our small-scale experiments and their frequency of identification (this frequency is expressed in qualitative terms owing to the small number of experiments performed so far). Across all students, the most frequent stakeholders identified on waterscapes were farmers, civil water supply, and industries, in line with most of the involved classes being located in Liguria (Italy), a region with a mixture of cities and rural areas. Hydropower and tourism were less frequently identified as water stakeholders, despite the latter being a key economical sector in Liguria. Across all stakeholders, ecosystems were the least frequent to be identified on our waterscapes. While frequency of identification will likely change with a larger sample, and in general depend on local knowledge and the experience of students, the fact that this stakeholder list tallies with that compiled by experts in the field is a

| Stakeholder | Frequency of identification (qualitative) | Comment |
|---|---|---|
| Farmers | HIGH | This category includes agriculture and breeding farms |
| Civil water supply | HIGH | - |
| Industries | HIGH | - |
| Hydropower | MEDIUM | - |
| Tourism | MEDIUM | Both summer- (beach resorts) and winter-driven (ski resorts) |
| Ecosystems | LOW | - |

**Table 2.** Main water stakeholders identified by students during our small-scale experiments and their qualitative frequency of identification.

promising result of the effectiveness of Water and Us to accomplish Objective #1 (e.g., see the USGS new water cycle diagram at https://www.usgs.gov/special-topics/water-science-school/science/water-cycle-diagrams, last access 26/08/2023).

To measure indicators of objective #2 ("educate students on the most salient aspects of climate change and its governance, including the difference between mitigation and adaptation, the role of international agreements, the scientific foundation of global warming, and how these processes can affect water availability at all scales – including future scenarios of water supply, floods, and droughts"), we propose submitting an informal questionnaire to students to gauge prior and a-posteriori awareness of students regarding climate change (see some examples in Table 1). Regarding a-posteriori awareness, the relative frequency

of keywords used by students in Module 2 during 2021/22 does show salient expressions related to this topic, including "climate change", "greenhouse gases", "emissions", "temperature", and "impacts" (Figure 6). Interestingly, students also used several policy-oriented words, such as "countries", "agreement", "security", "parties", "scenarios", and "conflicts". We propose to explicitly monitor through a questionnaire the prior and a-posteriori knowledge of terms emerged in Figure 6 as an indicator of student's awareness of the (often cumbersome) decision process characterizing adaptation and mitigation to climate change.

Achieving objective #3 represents the essential outcome of the whole process ("raise awareness on existing and potential governance conflicts around the use of water, especially in a warming climate, and on solutions for a nonconflictual water resources management."). Here again, the hands-on workshop provides concrete indicators to measure this gained awareness: we propose quantification through the average/variance number of identified synergies, conflicts, and "collaborators" across water stakeholders – as emerged across groups. Comparison across geographic locations as Water and Us progresses will allow

to draw a clear picture of how such tendency to synergies or conflicts change with time and space, especially as extremes emerge in a warming climate.

During our small-scale experiments in Liguria (Italy), two classes of potential conflicts emerged: the concurrent need of water (possibly different between summer and winter) and water quality preservation (Table 3). According to the involved students in Mediterranean Italy, all water stakeholders can be in conflict around the need of water in summer, with some

270 recurring patterns such as freshwater supply for residents vs. tourists, or irrigation water requirements vs. energy production. In our dry winters, the main source of conflict according to students is again around the concurrent need of water by residents vs. tourists (or tourist facilities in general, such as ski resorts). As for water quality, this is generally perceived as an "ecosystems

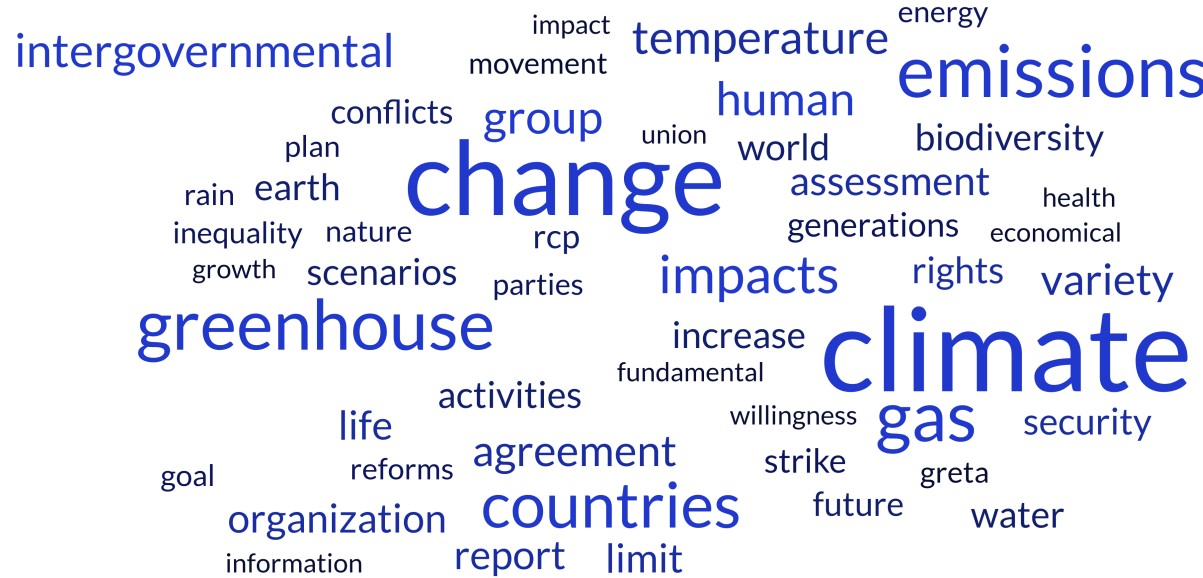

**Figure 6.** Relative frequency of keywords used by students in Module 2 (word cloud obtained with https://www.freewordcloudgenerator. com/, last access 27/08/2023).

vs. all" example of conflict due to the often elusive understanding of ecosystem preservation by the general public. The fact that students were able to spell out potential, concrete conflicts around the use of water, along with the fairly extensive list of

275 potential solutions (Table 3), speaks for the promising effectiveness of Module 3 to achieve Objective 3.

We finally propose a set of monitoring indicators to measure the overall impact of Water and Us in terms of involved students, teachers, and schools, as well as involved ages (see again Table 1). We also propose to monitor the number of students who take part to all modules and the number of schools asking for repetitions of Water and Us through school years as further measures of the impact of this program across years and across modules. Since Water and Us also represents an opportunity

for students to be exposed to the job market through the concrete topics discussed in the workshops, we finally propose to include as additional indicators the number of career-related follow-up questions by students, and importantly the number of schools that officially included Water and Us in their career-preparation portfolio. This was the case for one of our high-school events, which was contextualized in the so-called "Paths towards Cross-cutting Skills and Orientation" (in Italian, Percorsi per le Competenze Trasversali e per l'Orientamento – PCTO). Established at national level by law 145/2018, PCTOs promote

the development of soft and career-oriented skills in high-school students by exposing them to the job market in controlled environments. This framework did not modify the ambition and overall organization and spirit of Water and Us, but at the same time provided us with the opportunity to explicitly discuss with students about job opportunities related to climate change and science (Figure 7).

| Potential conflict | Involved stakeholders | Potential synergies/solutions |
|---|---|---|
| Concurrent need of water (summer) | All | Improve irrigation efficiency |
| | | Schedule day vs. night shifts |
| | | Sustainable tourism in farms and ecosystems |
| | | Improve distribution efficiency |
| | | Incentivize small hydro |
| | | Incentivize water-saving technologies |
| | | Improve touristic value of hydro reservoirs |
| | | Improve water reuse |
| Concurrent need of water (winter) | Civil water supply vs. Tourism (esp. in mountains) | Improve distribution efficiency |
| | | Improve snow-making efficiency |
| | | Incentivize water reuse |
| | | Incentivize water-saving technologies |
| Water-quality preservation | Ecosystems vs. all | Improve sanitation efficiency |
| | | Improve sanitation technologies |
| | | Raise awareness about ecosystem services |
| | | Encourage preservation by locals |

**Table 3.** Preliminary list of the most recurring classes of conflicts and solutions identified in Module 3.

Water and Us continued in 2022/23, nested in several national and European projects dedicated to climate-change awareness and communication. A concrete example in this regard is I-CHANGE (https://ichange-project.eu/, last access 06/09/2022), where Water and Us was part of educational activities related to the Living Lab paradigm in Genova (https://ichange-project.eu/open-air-laboratory-in-genoa/, last access 06/09/2022). Regarding future steps, we remain interested in networking with interested partners to scale up this experience at an international level. This will necessarily need ways to adapt Water and Us to different audiences and cultures. We identified three promising resources and one pre-condition in doing so. The first resource is the already mentioned Chapter 4 of the IPCC Assessment Report 6 on Water (see Section 2). The second resource is the framework provides by UN Water (https://www.unwater.org/, last access 20/05/2023), including its section on water facts providing concrete examples of water sustainability cases across the globe. The third resource is the UN Sustainable Development Goals (https://sdgs.un.org/goals, last access 20/05/2023), which also provide concrete examples of targets and metrics related to sustainability and climate change.

The main pre-condition in scaling up and transferring Water and Us is that these new editions should be collaboratively led by local scientists, rather than the present authors. This is important to maintain one of the most promising aspects of Water and Us storytelling and overall educational framework: it is about local students engaging with local scientists. The present authors remain available to accompany and support interested colleagues in this process.

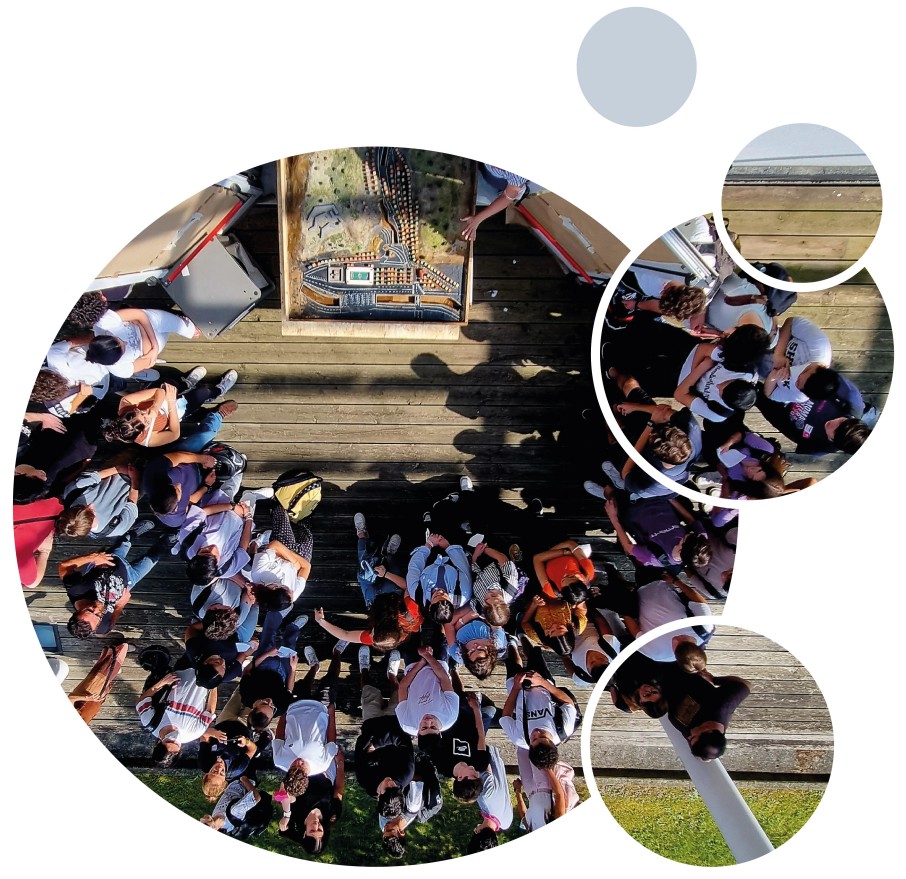

**Figure 7.** High-school students visiting CIMA Research Foundation to learn about floods during one of our Water and Us initiatives. This was an opportunity to discuss about employability in climate-change science as part of a "Paths towards Cross-cutting Skills and Orientation" (PCTO, see main text for details).

We also further aim to advance on concrete tools enabling students to make their voices heard in climate-change policy through Water and Us. This is important to improve the bottom-up aspect of this initiative and so allow students to inform our work as researchers in hydrology, policy, and governance. In order to achieve this goal, we are experimenting a fourth module of Water and Us where students propose concrete measures to tackle climate change, cluster in advocacy groups to promote their vision, and finally vote on each of these propositions (see https://www.cimafoundation.org/en/news/the-next-generation-cop/, last access 27/08/2023).

## 4 Conclusions

We presented Water and Us, an awareness initiative contributing to educating next generations in the challenges of water security and conflicts in a warming climate. Water and Us was established in 2021 and involved about 200 students and 100 adults across 40 hours of events in a first set of experiments in 3 schools to test the approach and validate it. We defined a repeatable structure for high schools made of three educational modules dedicated to the water cycle, climate change, and water conflicts. Water and Us affirms the value of storytelling and of learning by doing, while putting students at the center of a learning process made by hands-on workshops. We continued the experience of Water and Us in 2022/23, as part of EU Horizon projects geared towards behavioral change and education.

*Author contributions.* All coauthors contributed to the initial design of Water and Us. FM and FA developed the initial educational portfolio and tested it during the 2021-22 school year, with inputs from all coauthors. FM and FA prepared the first draft of this manuscript, with contributions from all coauthors.

*Competing interests.* Authors declare that no competing interests are present.

*Data availability.* An online repository of materials used in these workshops is available at: https://doi.org/10.5281/zenodo.8341482.

**Ethical statement**

The work presented is original, reflects the authors' observations, and does not deal with sensitive data. The work presented respects what was stated in the Helsinki Declaration of 1964, the cornerstone of the ethics of human research. Ethical approval was requested and obtained from the body to which the authors belong.

*Acknowledgements.* We express our sincere gratitude to all classes and schools who took part to Water and Us, comprising the ISS "Giovanni Falcone" in Loano (SV), the Liceo Scientifico Statale "Orazio Grassi" in Savona, and the Scuola Primaria "Perasso" in Genova. We also thank the "Prime Minister" initiative in Loano (SV, see https://www.primeminister.it/en/home-english/, last access 02/08/2022), the Rotary clubs Lovere-Iseo-Breno (BS), Albenga (SV), and Milano International, and the Trekking Nature initiative by the Aosta Valley Regional Administration. A special thanks goes to Antonio Parodi (CIMA Foundation) for fruitful discussions around Water and Us, Roberta Bellini (Trinity College, Dublin) for supporting our impact analysis, Marta Galvagno, Federico Grosso, and Edoardo Cremonese (Environmental Protection Agency of Aosta Valley) for supporting our activities in Aosta valley, Rita Visigalli and Giulia Cavallari (CIMA Foundation) for their support with figure preparation, and Tessa Maurer (Blue Forest, Sacramento) for her advice on the Appendix. We also acknowledge partial support from the H2020 I-CHANGE project (Grant Agreement no. 101037193).

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

**Appendix A: An example of storytelling – a tale of water and snow**

Winter 2021-22 hit the ground running in Italy, with the first snowflakes falling across the Alps in early November. Snow returned between November and early December, when a second large storm hit most Italian mountain ranges. Early snowfalls peaked on December 8, when snow reached sea level and covered many of our cities (thoughtfully doing so during a national holiday rather than on a busy workday!). After two winters of lockdown due to the COVID-19 pandemic, Italians were finally enjoying snow at its best.

Unfortunately, the season did not proceed as we hoped, and this wet start gave way to a prolonged, and similarly unusual, dry and warm period. Due to a persistent barrier of warm air on the western Mediterranean Sea (meteorologists call it a high-pressure ridge), almost no precipitation fell in northwestern Italy between mid-December and March, with only a couple of short storms in mid-February and mid-March providing limited relief. Meanwhile, warm, strong wind coming down off the Alps caused unseasonably high temperatures and largely melted the mountain snowpack. By the end of March, snow levels on the Italian Alps were 60% lower than the average of the previous 12 years (2010-2021, Figure A1)[1]. Water use is at its minimum during winter, so many Italians did not quite realize what was going on, and – importantly – what was about to unfold.

Spring and summer came like a wake-up call. Early loss of snow and the lack of rain quickly led to some of the lowest streamflow levels in recent history across the agricultural and industrial plains of the Po River (Figure A1). The river, a constant presence that many Italians respect and sometimes even fear during floods, was now just a slow, faint tickle, barely reaching its own outlet into the Adriatic Sea. Meanwhile, newspapers started using a word that many Italians were not prepared to hear, or handle: *drought*. With media coverage also came uncertainty and puzzlement, given how few of us were familiar with this creeping disaster: what happened to all that snow we started off with? What are we supposed to do now? How long will it last?

Then came emergency measures, like reducing irrigation water and releasing stored water from Alpine lakes[2]. In a country with millennia of fragmented history, these measures exacerbated endemic issues around who has the right to use water first and why. In our Mediterranean climate, the bulk of precipitation comes in fall through spring, meaning that drought conditions are likely to linger at least across summer and early fall. So here we are, in early July 2022, in somewhat uncharted waters, facing at least a few more months of drought[3].

And yet, what seems like uncharted waters for Italians is vivid and a growing reality in another part of the world. Between 2012 and 2016, California experienced a similarly intense snow and precipitation drought, caused by a high-pressure ridge sending storms north towards the Pacific Northwest rather than the Golden State (Californians called it the "Ridiculously Resilient Ridge"). Drought and low-snow conditions returned to the state in 2020 and show no signs of relenting – a new normal for the U.S.'s largest economy[4]. The full effects of the current dry period will not be known for years, but these certainly include a spike in tree mortality, a rise in wildfires, and expectedly severe water deficits. Events escalated in 2015,

---

[1] https://edo.jrc.ec.europa.eu/documents/news/GDO-EDODroughtNews202203_Northern_Italy.pdf, last access 04/09/2022

[2] https://www.adbpo.it/misure-definite-dallosservatorio-per-far-fronte-alla-crisi-idrica/, last access 04/09/2022

[3] https://edo.jrc.ec.europa.eu/documents/news/GDO-EDODroughtNews202208_Europe.pdf, last access 14/09/2022

[4] https://www.gov.ca.gov/2021/04/21/governor-newsom-takes-action-to-respond-to-drought-conditions/, last access 04/09/2022

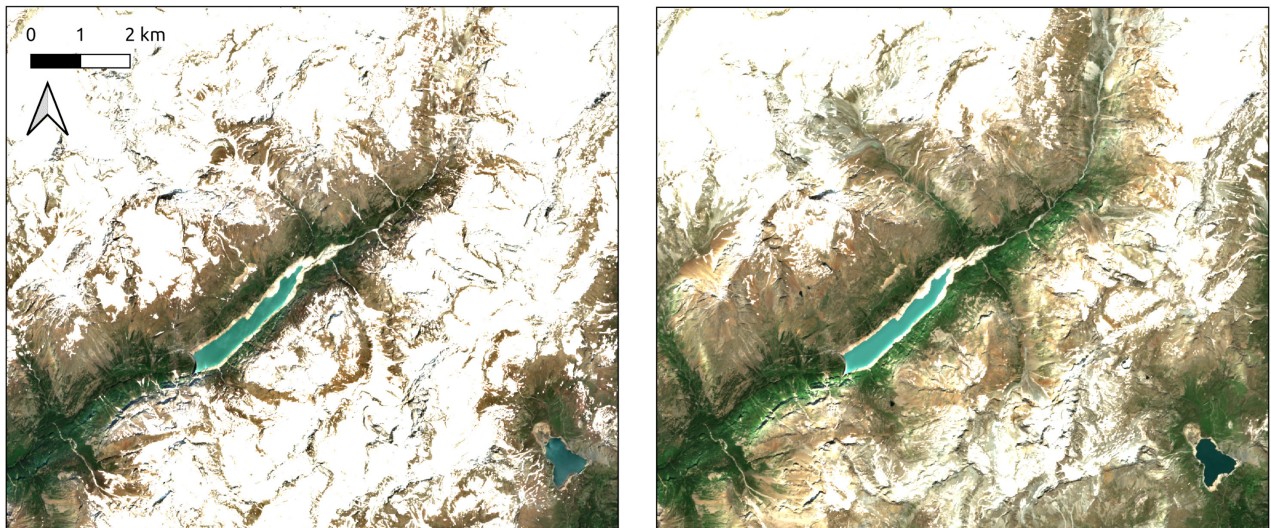

Place Moulin, June 2021 (left) vs. June 2022 (right) (c) ESA Copernicus

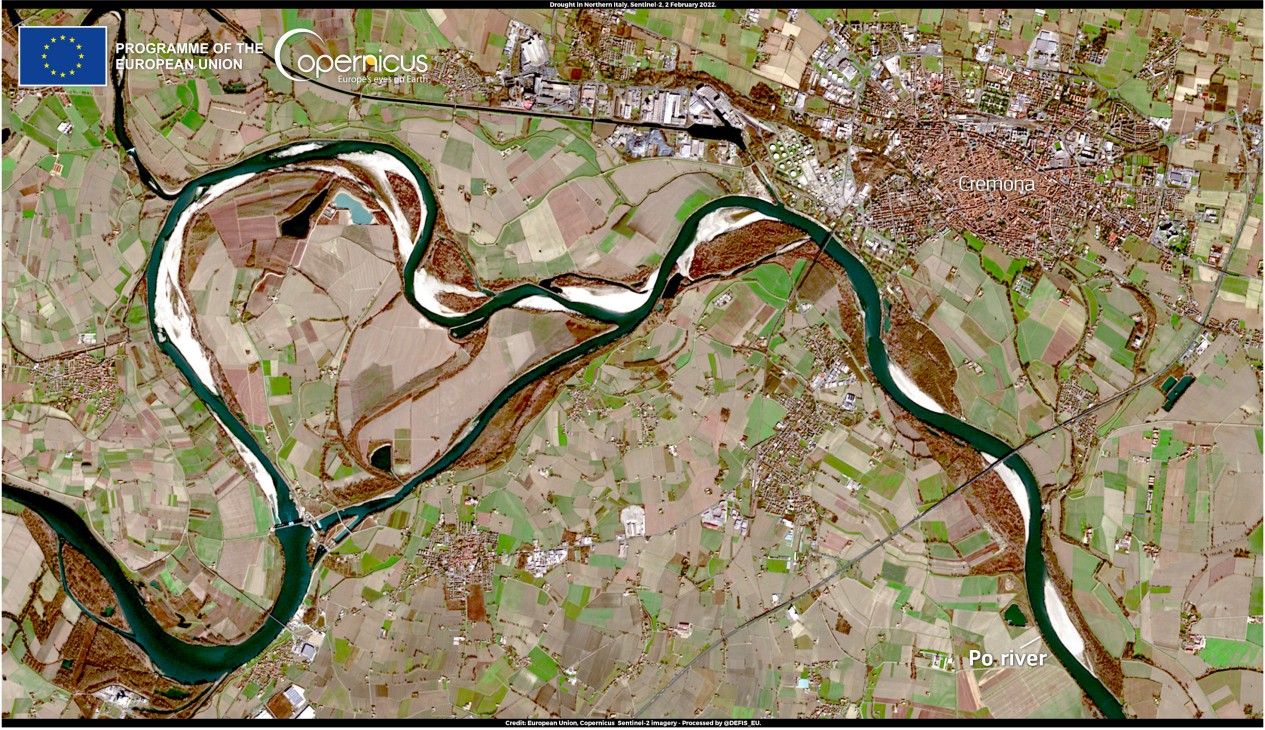

**Figure A1.** Two key features of the 2022 Italian drought, a marked deficit in snow cover (upper panel, Place Moulin in Aosta valley, June 2021 vs. 2022) and low streamflow (lower panel, Po river at Cremona). Credits: European Union, Copernicus Sentinel-2 imagery.

when the then governor – Mr. Jerry Brown – issued an executive order mandating a 25% reduction in water consumption across the state. As allocations across farmers, municipal users, ecosystems, and industries were becoming increasingly contentious, 440 California also passed landmark laws like the Sustainable Groundwater Management Act to protect groundwater from future non-sustainable use.

What California learned during the 2012–2016 drought is the same lesson that Italy is now learning the hard way – one that will characterize the whole 21st century: warmer temperatures (and occasionally less precipitation) could lead to less snow, less water, and ultimately more conflicts.