# Peer review of "Water and Us: tales and hands-on laboratories to educate on sustainable and nonconflictual water resources management"

_EGUsphere, 2022_

## Referee Comment (RC1)

Review 'Water and Us: tales and hands-on laboratories to educate on sustainable and nonconflictual water resources management'

General comments:

- The authors describe an interesting, interactive method of involving high school students in learning more about the natural and antrophogenic water cycle. The paper describing the project and first iteration, with 200+ students, is well written, bar some linguistic issues (see specific comments). I recommend this manuscript for publication in Geoscience Communication with minor revisions.
- The prologue could use a more descriptive title, adding in whether it was used in the method as part of the storytelling emphasis, or is added in here as background information, considering it is placed before the introduction. As an illustration of the importance of water, and the effects of climate change in a specific region, it is very effective, but more context as to how it fits into the project would be useful.
- All in all, the methodology and project description are very sound. It is obvious the authors have put much though into developing Water and Us, and its aims and set-up are compelling. As outlined below, section 4 needs work integrating the lessons learned into scientific context, but it is otherwise well-rounded.

Specific comments:

- L3: to contribute **to** advancing education
- L5: revolves instead of resolves
- L40: add 'a' before reality
- L45: 'precipitated' does not fit here. Depending on what the authors want to convey, use 'began' or other word
- L58: change 'by' to 'from'
- L62: would be good to have a more recent reference, to include scientific and societal development over the past 20 years
- Good recurring metaphor of the elephant – works very well here
- L71: add more recent reference, e.g. Immerzeel et al. (2020)
- L77: remainS anchored in, not anchored to
- L82: the authors here take the words by Kirsten von Elverfeldt out of context, in my opinion, saying climate change is an implausible risk. Von Elverfeldt argues that climate change *seems* implausible to non-science aligned people, not that it is. That should be clarified.
- L84: change 'for example' to also
- L88: educating THE next
- L94: please define 'high school students', as it is an ambiguous term, and very dependent on the country. A definition of age and/or subject it is tied to (as many high school students have a set directional curriculum) would be good.
- L108: I'm assuming the authors mean role playing games, rather than role games?
- The second pillar could use more elaboration: e.g. which roles are played in the role playing games? Those of stakeholders, affected people, decision makers, or all three? Especially compared to the description of the other pillars, this seems very short and lacks necessary detail
- L125: intuition is a peculiar choice of words here. I would suggest 'builds on the premise'

- L136: archetype is, again, a word that doesn't quite fit. I would suggest 'case study' or simply 'example'.
- L141: parallels instead of parallelisms
- L141: concurrent is not the right word here, because these events happened at different points in time. I suggest omitting it
- L145: 'laboratory' in English refers to a building rather than an activity. Please revise
- L157: see comment L145
- L172: take part IN
- L174: 'may apparently be challenging' is strangely worded. Please revise. I suggest simply 'is challenging'
- L174: associate WITH
- L175: replace 'regards' with 'conveys'
- L188: remove 'are the'
- L191: 'hammer down' is not correct. I suggest 'in order to both integrate'
- L192: 'proximal' is not the correct word here. I suggest simply 'close'
- Section 2.5 is very clear and well-written, I am very impressed
- L222: inTO
- L238: replace 'it strikes' with 'it is striking'
- L238: add 'the' to IPCC
- L241: I'm not sure 'diffusion' is the right word here, as it implies randomness. I personally (but of course the authors can disagree) would prefer 'dissemination' or 'transfer'.
- L242: 'has been' is the wrong tense to use here, replace with 'were'
- L248: replace 'on' with 'in'
- L252: 'suggestive' is not the right word here. Please revise
- L266: replace 'in' with 'of'
- L268: 'youngsters' too colloquial
- L281: change 'make' to 'making'
- Overall, I find section 4 lacking depth, as it tries to very broadly cover the lessons learned, and the authors attempt to connect their personal findings to previously done research and its conclusions. An example is the 'breakdown of youngsters' by Kuthe et al. (2019), which is applied, then not further addresses in its methodology of application to the Water and Us students, the ratios within the groups, or how these categorizations are helpful, and what specific needs each group has. This could have been tied into the second part of the section very well, which speaks about audience priorities. The categorizations could be used as groups to address different priorities. Finally, when discussing the future integration of Water and Us into future national and European projects, it would be good to know whether this will still be located in Italy, or whether the project will expand. If the latter, this would require at least some brief communication of how the project will then be (as expressed L279-281) tailored to the geographic location and socioeconomic circumstance.
- L308: change 'to' to 'in'
- L314: at AN international
- Overall, conclusion is very short, but to the point, and I think it fits here.

References:

Immerzeel, W. W., Lutz, A. F., Andrade, M., Bahl, A., Biemans, H., Bolch, T., ... & Baillie, J. E. M. (2020). Importance and vulnerability of the world's water towers. *Nature*, *577*(7790), 364-369.

---

## Author Comment (AC1)

**Response to R1, Larissa van der Laan**

**General comments:**

**The authors describe an interesting, interactive method of involving high school students in learning more about the natural and antrophogenic water cycle. The paper describing the project and first iteration, with 200+ students, is well written, bar some linguistic issues (see specific comments). I recommend this manuscript for publication in Geoscience Communication with minor revisions.**

> Thank you very much for your constructive feedback! Please find below a point-by-point reply to your comments and our intended changes to the manuscript.

**The prologue could use a more descriptive title, adding in whether it was used in the method as part of the storytelling emphasis, or is added in here as background information, considering it is placed before the introduction. As an illustration of the importance of water, and the effects of climate change in a specific region, it is very effective, but more context as to how it fits into the project would be useful.**

> We agree that more background is needed here. To this end, we will move this prologue after the methods, so that we will actively refer to it in the main body of the paper as an example of storytelling related to water and climate change. Thus, the necessary background will be provided.

**All in all, the methodology and project description are very sound. It is obvious the authors have put much though into developing Water and Us, and its aims and set-up are compelling. As outlined below, section 4 needs work integrating the lessons learned into scientific context, but it is otherwise well-rounded.**

> Thank you!

**Specific comments:**

**- L3: to contribute to advancing education**

**- L5: revolves instead of resolves**

**- L40: add 'a' before reality**

**- L45: 'precipitated' does not fit here. Depending on what the authors want to convey, use 'began' or other word**

**- L58: change 'by' to 'from'**

Noted. We will fix all the above in the revised manuscript.

**- L62: would be good to have a more recent reference, to include scientific and societal development over the past 20 years**

Noted. We will add a more updated reference.

**- Good recurring metaphor of the elephant – works very well here**

**- L71: add more recent reference, e.g. Immerzeel et al. (2020)**

**- L77: remainS anchored in, not anchored to**

Noted. We will fix all the above in the revised manuscript.

**- L82: the authors here take the words by Kirsten von Elverfeldt out of context, in my opinion, saying climate change is an implausible risk. Von Elverfeldt argues that climate change seems implausible to non-science aligned people, not that it is. That should be clarified.**

We agree and we will clarify this in the revised manuscript.

**- L84: change 'for example' to also**

**- L88: educating THE next**

Noted. We will fix all the above in the revised manuscript.

**- L94: please define 'high school students', as it is an ambiguous term, and very dependent on the country. A definition of age and/or subject it is tied to (as many high school students have a set directional curriculum) would be good.**

We agree. In Italy, high-school students are generally between 14 and 19 years old, while elementary-school students are between 6 and 11 years old. Water and Us is not tied to a specific subject or directional curriculum. We will clarify this in the revised manuscript.

**- L108: I'm assuming the authors mean role playing games, rather than role games?**

Correct. We will fix this.

**- The second pillar could use more elaboration: e.g. which roles are played in the role playing games? Those of stakeholders, affected people, decision makers, or all three? Especially compared to the**

**description of the other pillars, this seems very short and lacks necessary detail**

> We agree and we will clarify this in the revised manuscript. Students play the role of various water stakeholders (agriculture, ecosystem services, tourism, industry, etc). They formulate their needs about water and then compare needs across stakeholders to identify potential synergies and emerging conflicts.

**- L125: intuition is a peculiar choice of words here. I would suggest 'builds on the Premise'**

**- L136: archetype is, again, a word that doesn't quite fit. I would suggest 'case study' or simply 'example'.**

**- L141: parallels instead of parallelisms**

**- L141: concurrent is not the right word here, because these events happened at different points in time. I suggest omitting it**

**- L145: 'laboratory' in English refers to a building rather than an activity. Please revise**

**- L157: see comment L145**

**- L172: take part IN**

**- L174: 'may apparently be challenging' is strangely worded. Please revise. I suggest simply 'is challenging'**

**- L174: associate WITH**

**- L175: replace 'regards' with 'conveys'**

**- L188: remove 'are the'**

**- L191: 'hammer down' is not correct. I suggest 'in order to both integrate'**

**- L192: 'proximal' is not the correct word here. I suggest simply 'close'**

**- Section 2.5 is very clear and well-written, I am very impressed**

**- L222: inTO**

**- L238: replace 'it strikes' with 'it is striking'**

**- L238: add 'the' to IPCC**

**- L241: I'm not sure 'diffusion' is the right word here, as it implies randomness. I personally (but of course the authors can disagree) would prefer 'dissemination' or 'transfer'.**

**- L242: 'has been' is the wrong tense to use here, replace with 'were'**

**- L248: replace 'on' with 'in'**

**- L252: 'suggestive' is not the right word here. Please revise**

**- L266: replace 'in' with 'of'**

**- L268: 'youngsters' too colloquial**

**- L281: change 'make' to 'making'**

     Noted. We will fix all the above in the revised manuscript.

**- Overall, I find section 4 lacking depth, as it tries to very broadly cover the lessons learned, and the authors attempt to connect their personal findings to previously done research and its conclusions. An example is the 'breakdown of youngsters' by Kuthe et al. (2019), which is applied, then not further addresses in its methodology of application to the Water and Us students, the ratios within the groups, or how these categorizations are helpful, and what specific needs each group has. This could have been tied into the second part of the section very well, which speaks about audience priorities. The categorizations could be used as groups to address different priorities. Finally, when discussing the future integration of Water and Us into future national and European projects, it would be good to know whether this will still be located in Italy, or whether the project will expand. If the latter, this would require at least some brief communication of how the project will then be (as expressed L279-281) tailored to the geographic location and socioeconomic circumstance.**

     This section will undergo major revisions according to comment from you and reviewer #2. It will likely be merged with the Impact section and heavily summarised.

**- L308: change 'to' to 'in'**

**- L314: at AN international**

**- Overall, conclusion is very short, but to the point, and I think it fits here.**

     Noted. We will fix all the above in the revised manuscript.

---

## Author Comment (AC2)

**Response to R2, Maurits Ertsen**

**With apologies for being late with my comments, I was (and still am) very happy to see that a paper combining education, societal issues and water is being shared with the scientific community. Indeed, I agree with the authors that education is useful. What I am a little concerned about – or put differently: what I would like the authors to elaborate on in a next version of the text – are three issues. I will discuss these below, after which I provide a few remarks on specific elements in the text.**

Thank you very much for your constructive feedback! Please find below a point-by-point reply to your comments and our intended changes to the manuscript.

**Issue 1: methodological strength**

**Interesting as the paper is, I do not think that the material that is available allows for any assessment of the Water and Us design and method now. It is very difficult to see which data are actually being mobilized to support any of the claims about why Water and Us delivers on its promises. I'll discuss the potential challenge that WaU promises quite a few things in the other two Issues. For the methodological issue, most claims are supported by observations (which are occasionally labelled as "qualitative" I have to assume (eg 266 and 268)). I could not detect any clear description on how data to evaluate the effects of WaU are (to be) collected. The breakdown into impacts might work, but it would be needed to specify for each (rather different) category how data are to be found and applied in any analysis. Discussing the EC framework does not much more than mentioning what expectations the authors have (eg note the "potential" on line 251). The methods and underpinning evidence are quite anecdotal. Some numbers are merely descriptive, but how to understand them? As a small remark: the figures are not very informative, as they show only very small bits and pieces. As a side note on method: the claim that the didactic approach is sound is not too strange, but is also relatively thinly supported.**

Thanks for your feedback. We agree that the current state of Water and Us is a first step, rather than an already established methodology. Thus, we will address your concern in two ways:

1. We will elaborate on an array of objective indicators to be discussed in Section 3 and applied in future steps of Water and Us to measure impact and validate our educational hypotheses. We will comment on how our first experiences informed the definition of such objective indicators;
2. Following this and other comments of yours, we will edit our language to clarify that we are documenting our first step, which others can co-develop and further expand.

**Issue 2: role of education**

**As already mentioned above, claiming that education is useful is not that weird. It is also perfectly ok to suggest that education is something beyond learning how to do calculus or the like. I am also not surprised that current education is not using the "latest knowledge" as such. That is to be expected in systems that are slow, and that use material with a certain slowness in production adaptation. As such, using the type of workshop (laboratory) that the**

**text presents is actually a pretty good answer, as it would allow bringing in recent ideas in a flexible way into existing programs. However, in order to discuss whether the workshop WaU is effective, we would need to know how it links to school programs and approaches (which are expected to be different). One workshop in a sea of otherness might not change too much? My other concern is the rather automatic assumption that educating people results in better actions. I refer to Issue 3 for some remarks on "better actions" as such, and would like to suggest under Issue 2 that the relation between "education" and "action" is not straightforward at all. Knowing things does not mean that actions follow, either because there is no agreement or because one cannot take action. Furthermore, I do think that we have seen quite a few well-educated people doing rather undesirable things in history. Knowledge is political, action is (perhaps even more clearly) a political choice.**

We totally agree with you on this.

Regarding the link between Water and Us and current school programs, we will add one paragraph discussing how this initiative is closely linked to civics (in Italian, Educazione Civica, see https://www.istruzione.it/educazione_civica/) and to science programs in high school. Civics programs in Italy specifically include educational targets on sustainability and environment, while science programs cover topics related to Earth science, the water cycle, and climate. In this regard, please note that our workshops are closely designed with teachers, who actively take part in preparing the class and gathering feedback in the immediate aftermath.

Regarding your very interesting reflection on the link between education and action, we will add a discussion and some references in our revised manuscript on this. We will make sure no automatic link between education and action emerges from the text, and we will discuss best practices on how to connect these two aspects. We will also mention and elaborate on several EU projects geared towards behavioural change that are currently underway and supporting Water and Us (e.g., https://ichange-project.eu/).

**Issue 3: the complexity of the issue**

**This observation on knowledge and actions as political brings me to my final concern. I am quite sure that the designers of WaU are not aiming for a positivistic approach to climate, water and society. Having said that, the text does suggest quite clearly that there are good and bad explanations on topics, or that knowledge leads to defining solutions or avoiding conflicts. As soon as one allows stakeholders in (which WaU does, great!), I would suggest that one has to allow for different representations of "climate, water and society", or at least different claims on what is at stake and what needs to be done. And: whose story is told? Whereas the California drought – and the recent drought in Italy – are excellent entries into the complexity of the issue of "drought", the two examples provided in the text to show the importance of socio-hydrological focus (Dust Bowl and Maya) are simply not as straightforward as the text suggests. It is actually quite unclear how Maya society responded to drought – assuming that a society is a useful unit of analysis to discuss responses – if only because the evidence one uses matters quite a bit. This issue refers also to the cases and type of materials that are used in WaU: new evidence is coming in regularly, which can shift interpretations, but it could also be a case of different interpretations on the same evidence. The suggested relation between climate change, water cycle and conflict is actually not that straightforward.**

Thank you for these additional, valuable comments. We will both amend the example of storytelling (what is currently the prologue to our manuscript) to remove the Maya events, and

generally revise the text to avoid any positivistic or unidirectional approach to the topic of water and climate.

**Summary**

**In summary, I think the paper claims too much on methods and evidence on the Water and Us project, on the role of education in creating change, and on the topic of water, climate and society itself. What I could imagine, and would welcome very much, is that the paper invites others to try out the Water and Us approach. As such, publishing the experience so far would be a very good thing. It would mean for me, however, that the paper should quite drastically be changed in tone – with much less claiming and much more information on the process the module does in class. Such an invitation would also benefit from a much clearer designed methodology to evaluate the impact of the approach.**

We agree with this general overview. We will amend the text as outlined above and further elaborate on our methodology to be more precise on how Water and Us plays out in the classroom, as requested.

**Some remarks on text elements**

**Line 2: One would expect that high-school students miss certain knowledge, right, especially when it comes to larger, real-life issues?**

We agree and will clarify this.

**Line 11: Why use the term "fictious"? That does suggest there is also a "real" cycle?**

Yes, our experience is that the current understanding of the water cycle by students is based on a natural representation with no human interference. This is fictitious in essence, as the "real" water cycle does include human actions (as recently acknowledged by, e.g., the USGS: https://www.usgs.gov/special-topics/water-science-school/science/water-cycle-diagrams)

**Lines 13-14: This claim on education leading to less conflicts is too simple.**

We agree and will clarify this (see response to your general comments above).

**Lines 35-37: In many other countries issues on rights and access would arise too. Why use the term "endemic"?**

We agree that similar issues may arise elsewhere, and indeed this is the main idea behind Module 3 of Water and Us (see the reference to the Turkana Lake). However, we generally address an Italian audience (or, at least, this is our experience so far). As such, we think it is appropriate to keep a reference to Italy here. Throughout the text, we invite readers to elaborate on "local" examples that would be more relevant in other areas of the world.

The term "endemic" was metaphoric and meant that conflicts around the use of water have always existed in this country and are part of our everyday life.

**Line 50: The word "could" is very interesting here, as it could open up the whole question on what counts as knowledge, including evidence, uncertainties and representations.**

The word "could" was used here to merely denote the fact that climate change scenarios are still uncertain to some extent. We will clarify this.

**Line 53: Elephants in rooms tend to be invisible or at least made invisible. Is that an appropriate metaphor for climate change? There may be disagreement, especially on how to act, but I would not think climate change is invisible as an issue.**

We thank you for this comment. Our confusion might partially be because we are not native speakers. We meant that climate change is a topic that everyone knows about, but is often avoided in public discourse. Reviewer 1 appreciated this metaphor, which seems appropriate based on several online sources, so we would keep it in the text.

**Line 58: Why only mention one initiative?**

Thanks for this. If you refer to "Fridays for Future", this is by far the most well-known bottom-up initiative related to climate change based on our experience with Italian high-school students, which is why we used it here. We will add more initiatives in the revised manuscript.

**Lines 80-84: The many different remarks made show that a diversity of issues can be related to education. Which effect one accepts as more important (or more true…) might influence how one design educational formats. If complexity is the main reason for climate change being absent in teaching, one might come up with a different course compared to when issues are mixed up.**

Thanks for this. We will specify in the manuscript that Water and Us does start from the assumption that climate change is complex and contemporary, both aspects that make it difficult for teachers to fully cover it.

**Lines 85-92: Paragraph where many of the issues I refer to can be seen.**

We agree and will address them as outlined above.

**Lines 118-120: Does bringing in the ambiguity of policy and governance also refer back to the possible ambiguity of/in the (natural) sciences?**

**Line 120: Is "existing literature" one paper?**

Our narrative does include a constant reference to the uncertainty of climate change scenarios, and how decision makers take this uncertainty into account. At the same time, we are clear on what is currently known and understood, and how this knowledge is used to inform international agreements.

Regarding line 120, that was one example of the existing literature. We will add more.

**Paragraph 2.3: In Line 155, the claim is made that there is a clear vocabulary, whereas the remaining paragraph text suggest quite strongly that differences in definitions are real – which I think is actually very cool to show and to use in class. But how does this relate to the remark in Line 155? Does it mean that the authors argue that there is one set of correct definitions?**

We start from the assumption that a clear and precise vocabulary does exist (e.g., on what the Paris Agreement or greenhouse gases are – we largely rely on IPCC materials on this). At the same time, we also want students to understand that information that they might gather online or among themselves can be inaccurate, or simply partial. The second module of Water and Us aims at going from such incomplete definitions to precise ones. We also ask students to mention the source of information they used to come up with their proposed definition, so that we can comment on the reliability and accuracy of these sources.

**Line 175: Can one use the term "mismanagement"? Is the story that clear?**

We think that mismanagement is part of the problem, as we clarify with students (https://www.tandfonline.com/doi/abs/10.1080/14634980903578308?journalCode=uaem20). However, we also clarify that climate variability plays a role. This will be clarified.

**Lines 197-198: I do not agree that these three core messages can be directly associated with the three modules. These core messages at least use words/terms that were not too central in the descriptions of the modules.**

Of course, the link between an elementary-school version of Water and Us and its high-school counterpart is mediated by the need of changing lexicon and target. We will specify that the elementary-school version captures the component of Water and Us that are pertinent, relevant, and helpful to elementary school students.

**Line 208: The 70% is quite often used in discussions suggesting that water is important. I find it rather a cliché, but my more serious concern is that the body-water actually shows how complex the metaphor is: water is not visible at all in one's body, right? The body-type H20 is perhaps not the river-type H20?**

We agree this is a simplified concept, but we found it very effective with elementary-school children. Our experience is that they do understand this concept and it helps them familiarizing themselves with the importance of water.

**Line 220: Why is this framework useful or applicable?**

In this passage, we are shifting from methods to results. So we found it important to introduce how we moved from theory to practice. We will clarify this.

**Line 236: Is the 100% explained because it is a self-selected group? If so, the statistics is not terribly meaningful.**

The group of involved teachers was a mix between self-selected teachers and teachers who were involved on a later stage because Water and Us was already active in their school. In any case, we agree that this statistic should we interpreted with care. We will clarify this.

**Line 242: Using the term "diffusion" when it comes to knowledge goes quite against the idea of active learning, which would apply terms like "constructing knowledge".**

We will amend as suggested.

**Line 245-246: I see the link between what WaU does and the field of socio-hydrology, but I am not ready yet to accept the suggestion that a focus on education contributes to the scientific field of socio-hydrology as such. Perhaps this needs to be explained?**

We agree with this and will rephrase the passage accordingly.

**Line 266: What does "qualitatively high" mean here? I think we were informed that 90% had heard about the topic, but that would be "quantitatively high", right?**

We meant that our assessment was preliminary, and partially based on qualitative information. In this sense, the 90% awareness score is important, but should be subject to more extensive research in the future. We will clarify this.

**Lines 267-273: I would stay away from claims like this when one does not have more than some observations to back it up.**

This section will undergo major revisions according to comment from you and reviewer #1. In this sense, this section will likely be removed and heavily summarised in the Impact section.

**Line 273-275: We know that you argue such education is needed, and I would agree with that argument, but repeating this in a section on "lessons learned" or "future directions" seems a little strange to me.**

This section will undergo major revisions according to comment from you and reviewer #1. In this sense, this section will likely be removed and heavily summarised in the Impact section.

**Line 276: I find it quite shocking that finding out that different groups may need different approaches is presented as a result. I do appreciate mentioning it for sure, and do hope that the observation can be used as a design principle for education.**

This section will undergo major revisions according to comment from you and reviewer #1. In this sense, this section will likely be removed and heavily summarised in the Impact section.

**Lines 290-298: I find the issues of "local" and "global" fascinating, and have no real solution to overcome the divide – which may partially be artificial and is certainly political. I would be**

**interested to know more about the remark that the categories need different goals. Why would that be?**

This section will undergo major revisions according to comment from you and reviewer #1. In this sense, this section will likely be removed and heavily summarised in the Impact section.

**Line 293: Is "action" the same as "behavioural change"?**

In our view, action is a precondition for behavioural change. In any case, this section will undergo major revisions according to comment from you and reviewer #1. In this sense, this section will likely be removed and heavily summarised in the Impact section.

**Line 304: The idea that teaching Module 4 will "educate students to democracy and free speech" may be a little huge and optimistic? It does link to my earlier issues. I agree that education is linked to larger societal issues, but that does not mean that education can easily solve problems or bring improvements that easily. I think the idea that teaching the complexity of climate, water and society is already a challenge, and worthwhile in itself.**

We will revise wording as recommended and specify that the focus of Water and Us as it stands now is teaching the complexity of climate, water and society.

---

## Author Response (AR1)

**Response to R1, Larissa van der Laan**

**General comments:**

**The authors describe an interesting, interactive method of involving high school students in learning more about the natural and antrophogenic water cycle. The paper describing the project and first iteration, with 200+ students, is well written, bar some linguistic issues (see specific comments). I recommend this manuscript for publication in Geoscience Communication with minor revisions.**

> Public discussion: Thank you very much for your constructive feedback. Please find below a point-by-point reply to your comments and our intended changes to the manuscript.

**The prologue could use a more descriptive title, adding in whether it was used in the method as part of the storytelling emphasis, or is added in here as background information, considering it is placed before the introduction. As an illustration of the importance of water, and the effects of climate change in a specific region, it is very effective, but more context as to how it fits into the project would be useful.**

> Public discussion: We agree that more background is needed here. To this end, we will move this prologue after the methods, so that we will actively refer to it in the main body of the paper as an example of storytelling related to water and climate change. Thus, the necessary background will be provided.

> Changes to the manuscript: Done (prologue moved to Appendix and referred in the main text, where more background is provided).

**All in all, the methodology and project description are very sound. It is obvious the authors have put much though into developing Water and Us, and its aims and set-up are compelling. As outlined below, section 4 needs work integrating the lessons learned into scientific context, but it is otherwise well-rounded.**

> Public discussion: Thank you!

**Specific comments:**

**- L3: to contribute to advancing education**

**- L5: revolves instead of resolves**

**- L40: add 'a' before reality**

**- L45: 'precipitated' does not fit here. Depending on what the authors want to convey,**

**use 'began' or other word**

**- L58: change 'by' to 'from'**

> Public discussion: Noted. We will fix all the above in the revised manuscript.

> Changes to the manuscript: Done.

**- L62: would be good to have a more recent reference, to include scientific and societal development over the past 20 years**

> Public discussion: Noted. We will add a more updated reference.

> Changes to the manuscript: Done.

**- Good recurring metaphor of the elephant – works very well here**

**- L71: add more recent reference, e.g. Immerzeel et al. (2020)**

**- L77: remainS anchored in, not anchored to**

> Public discussion: Noted. We will fix all the above in the revised manuscript.

> Changes to the manuscript: Done.

**- L82: the authors here take the words by Kirsten von Elverfeldt out of context, in my opinion, saying climate change is an implausible risk. Von Elverfeldt argues that climate change seems implausible to non-science aligned people, not that it is. That should be clarified.**

> Public discussion: We agree and we will clarify this in the revised manuscript.

> Changes to the manuscript: This sentence was removed.

**- L84: change 'for example' to also**

**- L88: educating THE next**

> Public discussion: Noted. We will fix all the above in the revised manuscript.

> Changes to the manuscript: we welcomed the first comment, while the second comment refers to a sentence that was removed due to comments by Reviewer 2.

**- L94: please define 'high school students', as it is an ambiguous term, and very dependent on the country. A definition of age and/or subject it is tied to (as many high school students have a set directional curriculum) would be good.**

> Public discussion: We agree. In Italy, high-school students are generally between 14 and 19 years old, while elementary school students are between 6 and 11 years old. Water and Us is not tied to a specific subject or directional curriculum. We will clarify this in the revised manuscript.

> Changes to the manuscript: Done (see beginning of Section 2).

**- L108: I'm assuming the authors mean role playing games, rather than role games?**

> Public discussion: Correct. We will fix this.

> Changes to the manuscript: Done.

**- The second pillar could use more elaboration: e.g. which roles are played in the role playing games? Those of stakeholders, affected people, decision makers, or all three? Especially compared to the description of the other pillars, this seems very short and lacks necessary detail**

> Public discussion: We agree and we will clarify this in the revised manuscript. Students play the role of various stakeholders (agriculture, ecosystem services, tourism, industry, etc). They formulate their needs about water and then compare needs across stakeholders to identify potential synergies or emerging conflicts.

> Changes to the manuscript: Done (see lines 101ff)

**- L125: intuition is a peculiar choice of words here. I would suggest 'builds on the Premise'**

**- L136: archetype is, again, a word that doesn't quite fit. I would suggest 'case study' or simply 'example'.**

**- L141: parallels instead of parallelisms**

**- L141: concurrent is not the right word here, because these events happened at different points in time. I suggest omitting it**

**- L145: 'laboratory' in English refers to a building rather than an activity. Please revise**

**- L157: see comment L145**

**- L172: take part IN**

- L174: 'may apparently be challenging' is strangely worded. Please revise. I suggest simply 'is challenging'

- L174: associate WITH

- L175: replace 'regards' with 'conveys'

- L188: remove 'are the'

- L191: 'hammer down' is not correct. I suggest 'in order to both integrate'

- L192: 'proximal' is not the correct word here. I suggest simply 'close'

- Section 2.5 is very clear and well-written, I am very impressed

- L222: inTO

- L238: replace 'it strikes' with 'it is striking'

- L238: add 'the' to IPCC

- L241: I'm not sure 'diffusion' is the right word here, as it implies randomness. I personally (but of course the authors can disagree) would prefer 'dissemination' or 'transfer'.

- L242: 'has been' is the wrong tense to use here, replace with 'were'

- L248: replace 'on' with 'in'

- L252: 'suggestive' is not the right word here. Please revise

- L266: replace 'in' with 'of'

- L268: 'youngsters' too colloquial

- L281: change 'make' to 'making'

> Public discussion: Noted. We will fix all the above in the revised manuscript.

> Changes to the manuscript: Done.

- Overall, I find section 4 lacking depth, as it tries to very broadly cover the lessons learned, and the authors attempt to connect their personal findings to previously done research and its conclusions. An example is the 'breakdown of youngsters' by Kuthe et al. (2019), which is applied, then not further addresses in its methodology of application to the Water and Us students, the ratios within the groups, or how these categorizations are helpful, and what specific needs each group has. This could have been tied into the second part of the section very well, which speaks about audience priorities. The categorizations could be used as groups to address different priorities. Finally, when discussing the future integration of Water and Us into future national and European projects, it would be good to know whether this will still be located in Italy, or whether the project will expand. If the latter, this would require at least some brief communication of how the project will

**then be (as expressed L279-281) tailored to the geographic location and socioeconomic circumstance.**

> Public discussion: This section will undergo major revisions according to comment from you and reviewer #2. It will likely be merged with the Impact section and heavily summarised.

> Changes to the manuscript: Following comments by both reviewers, this section was removed and replaced by a more specific section on indicators and future steps to transfer Water and Us to other settings.

**- L308: change 'to' to 'in'**

**- L314: at AN international**

**- Overall, conclusion is very short, but to the point, and I think it fits here.**

> Public discussion: Noted. We will fix all the above in the revised manuscript.

> Changes to the manuscript: Done.

**Response to R2, Maurits Ertsen**

**With apologies for being late with my comments, I was (and still am) very happy to see that a paper combining education, societal issues and water is being shared with the scientific community. Indeed, I agree with the authors that education is useful. What I am a little concerned about – or put differently: what I would like the authors to elaborate on in a next version of the text – are three issues. I will discuss these below, after which I provide a few remarks on specific elements in the text.**

Public discussion: Thank you very much for your constructive feedback. Please find below a point-by-point reply to your comments and our intended changes to the manuscript.

**Issue 1: methodological strength**

**Interesting as the paper is, I do not think that the material that is available allows for any assessment of the Water and Us design and method now. It is very difficult to see which data are actually being mobilized to support any of the claims about why Water and Us delivers on its promises. I'll discuss the potential challenge that WaU promises quite a few things in the other two Issues. For the methodological issue, most claims are supported by observations (which are occasionally labelled as "qualitative" I have to assume (eg 266 and 268)). I could not detect any clear description on how data to evaluate the effects of WaU are (to be) collected. The breakdown into impacts might work, but it would be needed to specify for each (rather different) category how data are to be found and applied in any analysis. Discussing the EC framework does not much more than mentioning what expectations the authors have (eg note the "potential" on line 251). The methods and underpinning evidence are quite anecdotal. Some numbers are merely descriptive, but how to understand them? As a small remark: the figures are not very informative, as they show only very small bits and pieces. As a side note on method: the claim that the didactic approach is sound is not too strange, but is also relatively thinly supported.**

Public discussion: Thanks for your feedback. We agree that the current state of Water and Us is a first step towards a full-scale educational approach, rather than an already established methodology. Thus, we will address your concern in two ways:

1. We will elaborate on an array of objective indicators to be discussed in Section 3 and applied in future steps of Water and Us. We will comment on how our first experiences informed the definition of such indicators;
2. Following this and the other comments of yours, we will amend and edit our language to clarify that we are documenting a proof of concept that others are invited to co-develop and further expand.

Changes to the manuscript: We welcomed both suggestions. We replaced sections on impact and lessons learned with a new section on indicators (see Section 3). We agree that the current status of Water and Us is now much clearer. We also amended the text to clarify that we are documenting a proof of concept that others are invited to co-develop and further expand (see lines 14ff).

**Issue 2: role of education**

 **As already mentioned above, claiming that education is useful is not that weird. It is also perfectly ok to suggest that education is something beyond learning how to do calculus or the like. I am also not surprised that current education is not using the "latest knowledge" as such. That is to be expected in systems that are slow, and that use material with a certain slowness in production adaptation. As such, using the type of workshop (laboratory) that the text presents is actually a pretty good answer, as it would allow bringing in recent ideas in a flexible way into existing programs. However, in order to discuss whether the workshop WaU is effective, we would need to know how it links to school programs and approaches (which are expected to be different). One workshop in a sea of otherness might not change too much? My other concern is the rather automatic assumption that educating people results in better actions. I refer to Issue 3 for some remarks on "better actions" as such, and would like to suggest under Issue 2 that the relation between "education" and "action" is not straightforward at all. Knowing things does not mean that actions follow, either because there is no agreement or because one cannot take action. Furthermore, I do think that we have seen quite a few well-educated people doing rather undesirable things in history. Knowledge is political, action is (perhaps even more clearly) a political choice.**

Public discussion: We totally agree with you on this.

Regarding the link between Water and Us and current school programs, we will add one paragraph discussing how this initiative is closely linked to civics (in Italian, Educazione Civica, see https://www.istruzione.it/educazione_civica/) and to science programs in high school. Civics programs in Italy specifically include educational targets on sustainability and environment, while science programs cover topics related to Earth science, the water cycle, and climate. In this regard, please note that our workshops are closely designed with teachers, who actively take part in preparing the class and gathering feedback in the immediate aftermath.

Regarding your very interesting reflection on the link between education and action, we will add a discussion and some references in our revised manuscript on this. We will make sure no automatic assumption emerges from the text between education and action, and we will discuss best practices on how to link these two aspects (e.g., we are now working on a follow-up event of Water and Us where students take part to a simulation of a COP meeting and use the information obtained within Water and Us to identify concrete sustainability goals to pursue in their everyday life). We will also mention and elaborate on

several EU projects geared towards behavioural change that are currently underway and supporting Water and Us (e.g., https://ichange-project.eu/).

Changes to the manuscript: We welcomed both suggestions. We added one paragraph discussing the link between Water and Us and current school programs (see lines 64ff), while we revised wording to remove any automatic assumption between education and action. We also expanded our discussion of next steps in this regard in the new Section about indicators (see lines 275ff).

**Issue 3: the complexity of the issue**

**This observation on knowledge and actions as political brings me to my final concern. I am quite sure that the designers of WaU are not aiming for a positivistic approach to climate, water and society. Having said that, the text does suggest quite clearly that there are good and bad explanations on topics, or that knowledge leads to defining solutions or avoiding conflicts. As soon as one allows stakeholders in (which WaU does, great!), I would suggest that one has to allow for different representations of "climate, water and society", or at least different claims on what is at stake and what needs to be done. And: whose story is told? Whereas the California drought – and the recent drought in Italy – are excellent entries into the complexity of the issue of "drought", the two examples provided in the text to show the importance of socio-hydrological focus (Dust Bowl and Maya) are simply not as straightforward as the text suggests. It is actually quite unclear how Maya society responded to drought – assuming that a society is a useful unit of analysis to discuss responses – if only because the evidence one uses matters quite a bit. This issue refers also to the cases and type of materials that are used in WaU: new evidence is coming in regularly, which can shift interpretations, but it could also be a case of different interpretations on the same evidence. The suggested relation between climate change, water cycle and conflict is actually not that straightforward.**

Public discussion: Thank you for these additional, valuable comments. We will both amend the example of storytelling (what is currently the prologue to our manuscript) to remove the Maya events, and generally revise the text to avoid any positivistic or unidirectional approach to the topic of water and climate. On this matter, we will expand on how we usually stress the difference between *conflict* and *divergence*, and how knowledge might not necessarily lead to avoiding conflicts, but at least and having more tools to de-escalate such conflicts into divergences – which are often handled by state-of-the-art water resources management solutions.

Changes to the manuscript: We tackled this comment by removing any reference to Maya and by expanding on how to adapt Water and Us to other settings (see lines 275ff).

**Summary**

**In summary, I think the paper claims too much on methods and evidence on the Water and Us project, on the role of education in creating change, and on the topic of water, climate and society itself. What I could imagine, and would welcome very much, is that the paper invites others to try out the Water and Us approach. As such, publishing the experience so far would be a very good thing. It would mean for me, however, that the paper should quite drastically be changed in tone – with much less claiming and much more information on the process the module does in class. Such an invitation would also benefit from a much clearer designed methodology to evaluate the impact of the approach.**

Public discussion: We agree with this general overview. We will amend the text as outlined above and further elaborate on our methodology to be more precise on how Water and Us plays out in the classroom, as requested.

Changes to the manuscript: see above.

**Some remarks on text elements**

**Line 2: One would expect that high-school students miss certain knowledge, right, especially when it comes to larger, real-life issues?**

Public discussion: We agree and will clarify this.

Changes to the manuscript: given the small available space in the Abstract, we did not elaborate on this specific comment. At the same time, we did pay attention to avoid any positivistic approach to education.

**Line 11: Why use the term "fictious"? That does suggest there is also a "real" cycle?**

Public discussion: Yes, our experience is that the current understanding of the water cycle by students is based on a natural representation with no human interference. This is fictious in essence, as the "real" water cycle does include human action (as recently acknowledged by, e.g., the USGS: https://www.usgs.gov/special-topics/water-science-school/science/water-cycle-diagrams).

Changes to the manuscript: We changed this with "idealized".

**Lines 13-14: This claim on education leading to less conflicts is too simple.**

Public discussion: We agree and will clarify this (see response to your general comments above).

Changes to the manuscript: Revised.

**Lines 35-37: In many other countries issues on rights and access would arise too. Why use the term "endemic"?**

Public discussion: We agree that similar issues may arise elsewhere, and indeed this is the main idea behind Module 3 of Water and Us (see the reference to the Turkana Lake). However, we generally refer to an Italian audience (or, at least, this is our experience so far). As such, we think it is appropriate to keep this reference to Italy here – we are describing the Italian drought in the end. Throughout the text, we invite readers to elaborate on "local" examples that would be more relevant in other areas of the world.

The term "endemic" was metaphoric and meant that conflicts around the use of water have always existed in this country and are part of our everyday life. This will be amended with the word "historical" or similar.

Changes to the manuscript: We included passages on how to adapt Water and Us to other audiences and other settings (see lines 275ff).

**Line 50: The word "could" is very interesting here, as it could open up the whole question on what counts as knowledge, including evidence, uncertainties and representations.**

Public discussion: The word "could" was used here to merely denote the fact that climate change scenarios are still uncertain to some extent. We will clarify this.

Changes to the manuscript: No change, we think the intended meaning of this word is clear in this context.

**Line 53: Elephants in rooms tend to be invisible or at least made invisible. Is that an appropriate metaphor for climate change? There may be disagreement, especially on how to act, but I would not think climate change is invisible as an issue.**

Public discussion: We thank you for this comment. Our confusion might partially be because we are not native speakers. However, we meant that climate change is a topic that everyone knows about but is often avoided in public discourse because it can be controversial or divisive. Reviewer 1 appreciated this metaphor, which seems appropriate based on several online sources, so we would keep in the text.

Changes to the manuscript: No change.

**Line 58: Why only mention one initiative?**

Public discussion: Thanks for this. If you refer to "Fridays for Future", this is by far the most well-known bottom-up initiative – at least for our Italian audience, which is why we used it here. We will add more initiatives in the revised manuscript.

Changes to the manuscript: Done (see lines 21ff).

**Lines 80-84: The many different remarks made show that a diversity of issues can be related to education. Which effect one accepts as more important (or more true…) might influence how one design educational formats. If complexity is the main reason for climate change being absent in teaching, one might come up with a different course compared to when issues are mixed up.**

Public discussion: Thanks for this. We will specify in the manuscript that Water and Us does start from the assumption that climate change is complex and contemporary, both aspects that make it difficult for teachers to fully cover it.

Changes to the manuscript: This passage was removed in an effort to clarify our rationale (see lines 39ff).

**Lines 85-92: Paragraph where many of the issues I refer to can be seen.**

Public discussion: We agree and will address them as outlined above.

Changes to the manuscript: see previous comments.

**Lines 118-120: Does bringing in the ambiguity of policy and governance also refer back to the possible ambiguity of/in the (natural) sciences?**

**Line 120: Is "existing literature" one paper?**

Public discussion: Our narrative does include a constant reference to the uncertainty of climate change scenarios, and how decision makers take this uncertainty into account. At the same time, we are clear on what is currently known and understood, and how this knowledge is used to inform international agreements.

Regarding line 120, that was one example of the existing literature. We will add more.

Changes to the manuscript: Concerning line 120, we added "for example".

**Paragraph 2.3: In Line 155, the claim is made that there is a clear vocabulary, whereas the remaining paragraph text suggest quite strongly that differences in definitions are real – which I think is actually very cool to show and to use in class. But how does this relate to the remark in Line 155? Does it mean that the authors argue that there is one set of correct definitions?**

Public discussion: We start from the assumption that a clear and precise vocabulary does exist (e.g., on what the Paris Agreement is – we largely rely on IPCC materials on this). At the same time, we also want students to understand that information that they might gather online or among themselves can be inaccurate, or simply partial. The second module of Water and Us aims at going

from such incomplete definitions to precise ones. We also ask students to mention the source of information they used to come up with their proposed definition, so that we can comment on if and how such sources are reliable or not.

Changes to the manuscript: no change.

**Line 175: Can one use the term "mismanagement"? Is the story that clear?**

Public discussion: We think that mismanagement is part of the problem, as we clarify with students (https://www.tandfonline.com/doi/abs/10.1080/14634980903578308?journalCode=uaem20). However, we also clarify that climate variability plays a role. This will be clarified.

Changes to the manuscript: Word removed.

**Lines 197-198: I do not agree that these three core messages can be directly associated with the three modules. These core messages at least use words/terms that were not too central in the descriptions of the modules.**

Public discussion: Of course, the link between an elementary-school version of Water and Us and its high-school counterpart is mediated by the need of changing lexicon and target. We will specify that the elementary-school version captures the component of Water and Us that are pertinent, relevant, and helpful to elementary school students.

Changes to the manuscript: Done (see lines 201ff).

**Line 208: The 70% is quite often used in discussions suggesting that water is important. I find it rather a cliché, but my more serious concern is that the body-water actually shows how complex the metaphor is: water is not visible at all in one's body, right? The body-type H20 is perhaps not the river-type H20?**

Public discussion: We agree this is a simplified concept, but we found it very effective with elementary-school children. Our experience is that they do understand this concept and it helps them familiarizing themselves with the importance of water.

Changes to the manuscript: No change.

**Line 220: Why is this framework useful or applicable?**

Public discussion: In this passage, we are shifting from method description to results. So we found it important to introduce how we moved from theory to practise. We will clarify this.

> Changes to the manuscript: This section underwent major revisions, so the context should now be clearer (see lines 226ff).

**Line 236: Is the 100% explained because it is a self-selected group? If so, the statistics is not terribly meaningful.**

> Public discussion: The group of involved teachers was a mix between self-selected teachers and teachers who were involved on a later stage because Water and Us was already active in their school. In any case, we agree that this statistic should we interpreted with care. We will clarify this.

> Changes to the manuscript: This section underwent major revisions and this passage was removed.

**Line 242: Using the term "diffusion" when it comes to knowledge goes quite against the idea of active learning, which would apply terms like "constructing knowledge".**

> Public discussion: We took this wording from the European Commission's Horizon Europe breakdown in societal, scientific, and technological impacts. In any case, we will change with "constructing knowledge".

> Changes to the manuscript: This section underwent major revisions and this passage was removed.

**Line 245-246: I see the link between what WaU does and the field of socio-hydrology, but I am not ready yet to accept the suggestion that a focus on education contributes to the scientific field of socio-hydrology as such. Perhaps this needs to be explained?**

> Public discussion: We agree with this and will rephrase the passage accordingly.

> Changes to the manuscript: This section underwent major revisions and this passage was removed.

**Line 266: What does "qualitatively high" mean here? I think we were informed that 90% had heard about the topic, but that would be "quantitatively high", right?**

> Public discussion: We meant that our assessment was preliminary, and partially based on qualitative information. In this sense, the 90% awareness score is important, but should be subject to more extensive research in the future. We will clarify this.

> Changes to the manuscript: This section underwent major revisions, so the context should now be clearer (see lines 246ff).

**Lines 267-273: I would stay away from claims like this when one does not have more than some observations to back it up.**

Public discussion: This section will undergo major revisions according to comment from you and reviewer #1. In this sense, this section will likely be removed and heavily summarised in the Impact section.

Changes to the manuscript: This section underwent major revisions and this passage was removed.

**Line 273-275: We know that you argue such education is needed, and I would agree with that argument, but repeating this in a section on "lessons learned" or "future directions" seems a little strange to me.**

Public discussion: This section will undergo major revisions according to comment from you and reviewer #1. In this sense, this section will likely be removed and heavily summarised in the Impact section.

Changes to the manuscript: This section was removed.

**Line 276: I find it quite shocking that finding out that different groups may need different approaches is presented as a result. I do appreciate mentioning it for sure, and do hope that the observation can be used as a design principle for education.**

Public discussion: This section will undergo major revisions according to comment from you and reviewer #1. In this sense, this section will likely be removed and heavily summarised in the Impact section.

Changes to the manuscript: This section was removed.

**Lines 290-298: I find the issues of "local" and "global" fascinating, and have no real solution to overcome the divide – which may partially be artificial and is certainly political. I would be interested to know more about the remark that the categories need different goals. Why would that be?**

Public discussion: This section will undergo major revisions according to comment from you and reviewer #1. In this sense, this section will likely be removed and heavily summarised in the Impact section.

Changes to the manuscript: This section was removed.

**Line 293: Is "action" the same as "behavioural change"?**

Public discussion: In our view, action is a precondition for behavioural change. In any case, this section will undergo major revisions according to comment from you and reviewer #1. In this sense, this section will likely be removed and heavily summarised in the Impact section.

Changes to the manuscript: This section was removed.

**Line 304: The idea that teaching Module 4 will "educate students to democracy and free speech" may be a little huge and optimistic? It does link to my earlier issues. I agree that education is linked to larger societal issues, but that does not mean that education can easily solve problems or bring improvements that easily. I think the idea that teaching the complexity of climate, water and society is already a challenge, and worthwhile in itself.**

Public discussion: We will revise wording as recommended and specify that the focus of Water and Us as it stands now is teaching the complexity of climate, water and society.

Changes to the manuscript: We edited language here (see lines 290ff).

---

## Author Response (AR2)

Savona (Italy)

September 14, 2023

Dear Editors,

We would like to re-submit the manuscript *Water and Us: tales and hands-on laboratories to educate on sustainable and nonconflictual water resources management* for publication in Geoscience Communication.

The revised manuscript addresses the two remaining points raised by both the first Editor and the Executive Editor. Regarding the latter, we now clarified in the abstract (lines 3ff) and Introduction (lines 45ff) that our main goal here is to introduce the objective, methods, and early results of "Water and Us" in an effort to encourage other scientists and teachers to replicate this approach. This means that we are not aiming at testing storytelling in particular, although this remains a key component or our approach. To substantiate this goal, we expanded Section 3 to include some new results emerged from our first year of workshops. This section now includes concrete evidence of the effectiveness of Water and Us in achieving each of the three objectives as outlined in Section 2. In doing so, we did clarify that this evidence is preliminary, owing to the initial stage of this initiative (see e.g. line 245).

Regarding the remaining comments by the initial Editor, we have now initiated an online, open repository for the materials of this initiative (`https://doi.org/10.5281/zenodo.8341482`), which we will continue populating as possible and allowed by third-party copyright. We also addressed the remaining minor comments regarding language.

We hope that the manuscript is now suitable for publication and thank all of you for your efforts with this submission.

With our best regards,

*Francesca Munerol, Francesco Avanzi and coauthors*

---

## Author Response (AR3)

Savona (Italy)

December 8, 2023

Dear Editors,

We would like to re-submit the manuscript *Water and Us: tales and hands-on laboratories to educate on sustainable and nonconflictual water resources management* for publication in Geoscience Communication.

We included the recommended edit at line 62-63.

With our best regards,

*Francesca Munerol, Francesco Avanzi and coauthors*